

# Global drought and severe drought affected population in 1.5$^{o}$C and 2$^{o}$C warmer worlds

Wenbin Liu[a], Fubao Sun[a,b,c,d*], Wee Ho Lim[a,e], Jie Zhang[a], Hong Wang[a],

Hideo Shiogama[f] and Yuqing Zhang[g]

[a] Key Laboratory of Water Cycle and Related Land Surface Processes, Institute of Geographic Sciences and Natural Resources Research, Chinese Academy of Sciences, Beijing, China

[b] Ecology Institute of Qilian Mountain, Hexi University, Zhangye, China

[c] College of Resources and Environment, University of Chinese Academy of Sciences, Beijing, China

[d] Center for Water Resources Research, Chinese Academy of Sciences, Beijing, China

[e] Environmental Change Institute, University of Oxford, Oxford, UK

[f] Center for Global Environmental Research, National Institute for Environmental Studies, Tsukuba, Japan

[g] College of Atmospheric Sciences, Nanjing University of Information Science and Technology, Nanjing, China

**Pre-submit to**: Earth System Dynamics (*Special issue: Earth System at a 1.5$^{o}$C warming world*)

**Corresponding to**: Prof. Fubao Sun (*sunfb@igsnrr.ac.cn*), Institute of Geographic Sciences and Natural Resources Research, Chinese Academy of Sciences

2018/1/19

**Abstract.** The 2015 Paris Agreement proposed a more ambitious climate change mitigation target, on limiting the global warming at 1.5$^o$C instead of 2$^o$C above pre-industrial levels. Scientific investigations on environmental risks associated with these warming targets are necessary to inform climate policymaking. Based on the CMIP5 (the fifth Coupled Model Intercomparison Project) climate models, we present the first risk-based assessment of changes in global drought and the impact of severe drought on population at 1.5$^o$C and 2$^o$C additional warming conditions. Our results highlight the risk of drought at the globe (drought duration would increase from 2.9 to 3.1~3.2 months) and in several hotspot regions such as Amazon, Northeastern Brazil, South Africa and Central Europe at both 1.5 $^o$C and 2 $^o$C global warming relative to the historical period. Correspondingly, more total and urban population would be exposed to severe droughts at the globe (+132.5±216.2 million and +194.5±276.5 million total population, +350.2±158.8 million and +410.7±213.5 million urban population in 1.5 $^o$C and 2 $^o$C warmer worlds) and some regions (i.e., East Africa, West Africa and South Asia). Less rural population would be exposed to severe drought at the globe under both climate warming and population growth (especially the urbanization-induced population migration). By keeping global warming at 1.5$^o$C above the pre-industrial levels instead of 2$^o$C, drought risks would decrease (i.e., less drought duration, drought intensity and drought severity but relatively more frequent severe drought) and the affected total, urban and rural population would decrease at the globe and in most regions. Whilst challenging for both East Africa and South Asia, the benefits of limiting warming to below 1.5$^o$C in terms of global drought risk and impact reduction are significant.

**1 Introduction**

Drought could bring adverse consequences on water supplies, food productions and the
environment (Wang et al., 2011; Sheffield et al., 2012). Because of these serious consequences,
severe droughts in the recent past have gained wide attentions, these include the Millennium
drought in Southeast Australia (van Dijk et al., 2013; Kiem et al., 2016), the once-in-a-century
droughts in Southwest China (Qiu, 2010, Zuo et al., 2015), the Horn of Africa drought (Masih et
al., 2014; Lyon, 2014) and the most recent California drought (Aghakouchak et al., 2015; Cheng et
al., 2016). In the context of climate change, drought risks (i.e., drought duration and intensity) are
likely to increase in many historical drought-prone regions with global warming (Dai et al., 2012;
Fu and Feng, 2014; Kelley et al., 2015; Ault et al., 2016). A better understanding of changes in
global drought characteristics and their socioeconomic impacts in the 21st century should feed
into long-term climate adaptation and mitigation plans.

The United Nations Framework Convention on Climate Change (UNFCCC) agreed to establish a
long-term temperature goal for climate projection of "*pursue efforts to limit the temperature*
*increase to 1.5$^o$C above pre-industrial levels, recognizing that this would significantly reduce the*
*risks and impacts of climate change*" (UNFCCC Conference of the Parties, 2015) in the 2015 Paris
Agreement, and invited the Intergovernmental Panel on Climate Change (IPCC) to announce a
special report "*On the impacts of global warming of 1.5$^o$C above pre-industrial levels and related*
*greenhouse gas emission pathways*" in 2018 (Mitchell et al., 2016). Regardless of the
socio-economic and political achievability of this goals (Sanderson et al., 2017), there is a paucity
of scientific knowledge about the relative risks (i.e., drought risks and their potential impacts)
associated with the implications of 1.5$^o$C and/or 2$^o$C warming, this naturally attracted
contributions from scientific community (Hulme 2016, Schleussner et al., 2016, Peters 2016, King
et al., 2017).

To target on the impact assessments of 1.5$^{o}$C and/or 2$^{o}$C warming, there are currently several
approaches (James et al., 2017). One way is to enable impact assessments at near-equilibrium
climate of 1.5$^{o}$C and/or 2$^{o}$C warmer worlds designed specifically using a set of ensemble
simulations produced by a coupled climate model (i.e., Community Earth System Model, CESM)
(Sanderson et al., 2017; Wang et al., 2017). Although similar results of drought response to
warming were obtained as that conducted by Coupled Model Intercomparison Project-style
experiments (i.e., CMIP5, Taylor et al., 2012), the structural uncertainty and robustness of change
in droughts among different climate models cannot be fully evaluated in this kind of single-model
study (Lehner et al., 2017). A second approach extends the former idea to multiple climate
models. For instance, the HAPPI (Half a degree Additional warming, Projections, Prognosis and
Impacts) model intercomparison project provided a new assessment framework and a dataset
with experiment design target explicitly to 1.5$^{o}$C and 2$^{o}$C above the pre-industrial levels (Mitchell
et al., 2017). However, the analysis/calculation of drought characteristics needs data with
long-term period (typically >20 consecutive years, McKee et al., 1993), the ten-year period HAPPI
dataset (i.e., 2005-2016 for the historical period and 2105-2116 for the 1.5$^{o}$C and 2$^{o}$C warmer
worlds) is relatively short (consecutive samples are too short for calculating a drought index, i.e.,
Palmer drought severity index (PDSI), Palmer, 1965) for an index-based drought assessment. A
third approach utilizes the outputs of CMIP5 climate models under the RCP2.6 scenarios    for
this kind of "risk assessment-style" studies, but only a handful of General Circulation Models
(GCMs) simulations end up showing $1.5^oC$ global warming by end of the 21$^{st}$ century.
Alternatively, transient simulations from multiple CMIP5 GCMs at higher greenhouse emissions
(i.e., the RCP4.5 and RCP8.5) (Schleussner et al., 2016; King et al., 2017) could be analyzed in
order to evaluate the potential risks of drought under different warming targets, albeit the
long-duration drought years might be underestimated due to insufficient sampling of extended
drought events (Lehner et al., 2017).

Here, we quantify the changes in global and sub-continental drought characteristics (i.e., drought
duration, intensity and severity) at $1.5^oC$ and $2^oC$ above the pre-industrial levels and find out
whether there are significant differences between them. We perform this analysis using a
drought index-PDSI forced by the latest CMIP5 GCMs. To evaluate the societal impacts, we
incorporate the Shared Socioeconomic Pathway 1 (SSP1) spatial explicit global population
scenario and examine the exposure of population (including rural, urban and total population) to
severe droughts. This paper is organized as follows: Section 2 introduces the CMIP5 GCMs output
and SSP1 population data applied in this study. We define the baseline, $1.5^oC$ and $2^oC$ warmer
world; and describe the calculation of PDSI-based drought characteristics and population
exposure under severe droughts in this section. Section 3 shows the results (i.e., hotspots and
risks) of changes in drought characteristics and the impacts of severe drought on people under
these warming targets. We perform detailed discussions in Section 4 and conclude our findings in
Section 5.

**2 Material and Methods**

**2.1 Data**

In this study, we use the CMIP5 GCMs output (including the monthly outputs of surface mean air temperature, surface minimum air temperature, surface maximum air temperature, air pressure, precipitation, relatively humidity, surface downwelling longwave flux, surface downwelling shortwave flux, surface upwelling longwave flux and surface upwelling shortwave flux as well as the daily outputs of surface zonal velocity component ($uwnd$) and meridional velocity component ($vwnd$)) archived at the Earth System Grid Federation (ESGF) Node at the German Climate Computing Center-DKRZ (https://esgf-data.dkrz.de/projects/esgf-dkrz/) over the period 1850-2100. In the CMIP5 archive, the monthly $uwnd$ and $vwnd$ were computed as the means of their daily values with the plus-minus sign, the calculated wind speed from the monthly $uwnd$ and $vwnd$ would be equal to or, in most cases, less than that computed from the daily values (Liu and Sun, 2016). To get the monthly wind speed, we average the daily values ($\sqrt{uwnd^2 + vwnd^2}$) over a month.

Recent studies have confirmed that the impacts of similar global mean surface temperature (i.e., 1.5$^o$C and 2$^o$C warmer worlds) among the Representative Concentration Pathways (RCPs) are quite similar, implying that the global and regional responses to temperature and are independent of the RCPs (Hu et al., 2017; King et al., 2017). Following this idea, we settled at using 11 CMIP5 models which satisfied the data requirement of PDSI calculation (see paragraph above) under RCP4.5 and RCP8.5. Following Wang et al. (2017) and King et al. (2017), we use the ensemble mean of these CMIP5 models and climate scenarios (RCP4.5, RCP8.5) to composite the warming scenarios (1.5$^o$C and 2$^o$C warmer worlds).

<Table 1, here, thanks>

To consider the people affected by severe drought events, we use the spatial explicit global
population scenarios developed by researchers from the Integrated Assessment Modeling (IAM)
group of National Center for Atmospheric Research (NCAR) and the City University of New York
Institute for Demographic Research (Jones and O'Neil, 2016). They included the gridded
population data for the baseline year (2000) and for the period of 2010-2100 in ten-year steps at
a spatial resolution of 0.125 degree, which are consistent with the new Shared Socioeconomic
Pathways (SSPs). We apply the population data of the SSP1 scenario, which describes a future
pathway with sustainable development and low challenges for adaptation and mitigation. We
upscale this product to a spatial resolution of $0.5^o \times 0.5^o$. For the global and sub-continental scales
analysis, we use the global land mass between $66^o$N and $66^o$S (Fischer et al., 2013; Schleussner et
al., 2016) and 26 sub-continental regions (as used in IPCC, 2012, see Table 2 for details).

<Table 2, here, thanks>

**2.2 Definition of a baseline, 1.5$^o$C and 2$^o$C warmer worlds**
To define a baseline, 1.5$^o$C and 2$^o$C warmer worlds, we first calculate the global mean surface air
temperature (GMT) for each climate model and emission scenario over the period 1850-2100.
We weigh the surface air temperature field   by the square root of cosine (latitude) to consider
the dependence of grid density on latitude (Liu et al., 2016). We compute and smooth the
multi-model Ensemble Mean (MEM) GMT using a 20-year moving average filter for the RCP4.5
and RCP8.5, respectively. This study applied continuous time series for identification of drought
duration, intensity and severity. From the climate model projections, we noticed that
inter-annual variation of global mean air temperature is common and its magnitude differs with
different climate models. To account for it, following Wang et al. (2017), we first select a baseline
period of 1986-2005 for which the observed GMT was about 0.6$^o$C warmer (the MEM GMT was
0.4~0.8 $^o$C warmer during this period in 11 climate models used) than the pre-industrial levels
(1850-1900, IPCC, 2013). This is also a common reference period for climate impact assessment
(e.g., Schleussner et al., 2016). Next, for each emission scenario (RCP4.5 or RCP8.5), we define
the periods (Figure 1) during which the 20-year smoothed GMT increase by 1.3~1.7 $^o$C
(2027-2038 under the RCP4.5 and 2029-2047 under the RCP 8.5) and by 1.8~2.2 $^o$C (2053-2081
under the RCP4.5 and 2042-2053 under the RCP 8.5) above the pre-industrial period as the 1.5$^o$C
and 2$^o$C warmer worlds, respectively (as in King et al., 2017). To reduce the projection uncertainty
inherited from different emission scenarios, we combine (by averaging) the results of drought
characteristics and population exposures calculated for selected periods under the RCP4.5 and
RCP8.5, to represent the ensemble means of drought risk at 1.5$^o$C or 2$^o$C warmer worlds. In the
1.5$^o$C and 2$^o$C warmer worlds, we get 372 and 492 monthly data-points, respectively.

<Figure 1, here, thanks>

**2.3 Characterize global drought using PDSI**
To quantify the changes in drought characteristics, we adopt the Palmer Drought Severity index
(PDSI), which describes the balance between water supply (precipitation) and atmospheric
evaporative demand (required "precipitation" estimated under climatically appropriate for
existing conditions, CAFEC) at the monthly scale (Wells et al., 2004; Zhang et al., 2016). For a
multiyear time series, this index is commonly applied as an indication of a meteorological
drought; to a lesser extent, a hydrological drought (Heim Jr., 2002; Zargar et al., 2011; Hao et al.,
2018). It incorporates antecedent precipitation, potential evaporation and the local Available
Water Content (AWC, links: https://daac.ornl.gov/cgi-bin/dsviewer.pl?ds_id=548) of the soil in
the hydrological accounting system. It measures the cumulative departure relative to the local
mean conditions in atmospheric moisture supply and demand on land surface. In the PDSI model,
five surface water fluxes, namely, precipitation ($P$), recharge to soil ($R$), actual evapotranspiration
($E$), runoff ($RO$) and water loss to the soil layers ($L$); and their potential values $\hat{P}$, $PR$, $PE$, $PRO$ and
$PL$ are considered. All values in the model can be computed under CAFEC values using the
precipitation, potential evaporation and AWC inputs. For example, the CAFEC precipitation ($\hat{P}$) is
defined as (Dai et al., 2011),
$$\hat{P} = \frac{\overline{E_t}}{\overline{PE_t}}PE + \frac{\overline{R_t}}{\overline{PR_t}}PR + \frac{\overline{RO_t}}{\overline{PRO_t}}PRO - \frac{\overline{L_t}}{\overline{PL_t}}PL,\qquad\qquad (1)$$
In Equation 1, the over bar indicates averaging of a parameter over the calibration period. The
moisture anomaly index ($Z$ index) is derived as the product of the monthly moisture departure
($P - \hat{P}$) and a climate characteristics coefficient $K$. The $Z$ index is then applied to calculate the
PDSI value for time $t(X_t)$:
$$X_t = pX_{t-1} + qZ_t = 0.897X_{t-1} + Z_t/3 \qquad\qquad (2)$$
Where $X_{t-1}$ is the PDSI of the previous month, and $p$ and $q$ are duration factors. The
calculated PDSI ranges -10 (dry) to 10 (wet). The parameters (i.e., the duration factor) in PDSI
model are calibrated using the period 1850-2000 (see Section 2.2).

As part of the PDSI calculation, we quantify the potential evaporation ($PET$) using the Food and
Agricultural Organization (FAO) Penman-Monteith equation (Allen et al., 1998),
$$PET = \frac{0.408\Delta(R_n - G) + \gamma\frac{900}{T+273}U_2 e_s(1-\frac{Rh}{100})}{\Delta + \gamma(1+0.34U_2)} \qquad\qquad (3)$$
where $\Delta$ is the slope of the vapor pressure curve, $U_2$ is the wind speed at 2 m height, $G$ is the
soil heat flux, $Rh$ is the relative humidity, $\gamma$ is the psychometric constant, $e_s$ is the saturation
vapor pressure at a given air temperature ($T$). $R_n$ is the net radiation which can be calculated
using the surface downwelling/upwelling shortwave and longwave radiations. We estimate all
other parameters in the FAO Penman-Monteith equation using the GCM outputs through the
standard algorithm as per recommended by the FAO (Allen et al., 1998). In this study, we perform
this calculation for each GCM over the period 1850-2100 using the tool for calculating the PDSI
(the original MATLAB codes were modified for this case) developed by Jacobi et al. (2013).

Based on the calculated global PDSI, we derive the drought characteristics (i.e., drought duration,
drought intensity and drought severity) using the run theory for the baseline, 1.5$^{o}$C and 2$^{o}$C
warmer worlds, respectively. Briefly, the concept of "run theory" is proposed by Yevjevich and
Ingenieur (1967). The run characterizes the statistical properties of sequences in both time and
space. It is useful for defining drought in an objective manner. In the run theory, a "run"
represents a portion of time series $X_i$, where all values are either below or above a specified
threshold (we set the threshold PDSI < -1 in this study) (Ayantobo et al., 2017). We define a "run"
with values continuously stay below that threshold (i.e., negative run) as a drought event, which
generally includes these characteristics: drought duration, drought intensity and drought severity
(see Figure 2 for better illustration). We define the drought duration (months in this study) as a
period (years/months/weeks) which PDSI stays below a specific threshold (PDSI < -1). Drought
severity (dimensionless) indicates a cumulative deficiency of a drought event below the threshold
(PDSI < -1), while drought intensity (dimensionless) is the average value of a drought event below
this threshold (Mishra and Singh, 2010). For each GCM, we calculate the medians of drought
duration, drought intensity and drought severity at each grid-cell across all drought events for
each selected period (i.e., the baseline, 1.5$^{o}$C and 2$^{o}$C warmer worlds). It should be noted that
the global PDSI and related drought characteristics were first calculated using GCM-outputs with
their original spatial resolution. The obtained results were then rescaled to a common spatial
resolution of 0.5$^{o}$ ×0.5$^{o}$ using the bilinear interpolation, in order to show them with a finer
resolution uniformly and accommodate their spatial resolution to that of SSP1 population (0.5$^{o}$
×0.5$^{o}$). The original resolution of SSP1 population is 0.125 degree. We thus use a 0.5 degree
resolution to avoid effectively making up data of the finer resolution in SSP1 data. We synthesize
the results by evaluating the ensemble mean and model consistency/inter-model variance across
all climate models.

<Figure 2, here, thanks>

**2.4 Calculation of population exposure under severe droughts**
Following Wells et al. (2014), when monthly PDSI < -3, we assume a severe drought event took
place. If a severe drought occurred for at least a month in a year, we would take that year as a
severe drought year. For each GCM per period (i.e., the baseline, 1.5$^{o}$C and 2$^{o}$C warmer worlds),
we quantify the population (including urban, rural and all population) affected by severe drought
per grid-cell as (population × annual frequency of severe drought). We first compute the
affected population for the baseline period (1985-2005) using the SSP1 base year (2000). We
repeat this estimation using the constant SSP1 population data in 2100 for the 1.5$^{o}$C and 2$^{o}$C
warming worlds, which is consistent with the original proposal of Paris Agreement on stabilizing
global warming for the specified targets by end of the 21$^{st}$ century. We used SSP1 scenario

245 because it describes the storyline of a green growth paradigm with sustainable development and

246 low challenges for adaptation and mitigation (Jones and O'Niell, 2016). The 1.5$^o$C and 2$^o$C

247 warmer worlds clearly fit in this description and thus considered under the 2015 Paris Agreement

248 (UNFCCC Conference of the Parties 2015; O'Niell et al., 2016). In this pathway, the world

249 population would peak at around 2050s and then decline (van Vuuren et al., 2017). The

250 environmental friendly living arrangements and human settlement design in this scenario would

251 lead to fast urbanization in all countries. More in-migrants from rural areas would be attracted to

252 cities due to more adequate infrastructure, employment opportunities and convenient services

253 for their residents (Cuaresma, 2012). The world urban population would gradually increase while

254 rural population would correspondingly decline in the future under SSP1 scenario.

256 **3 Results**

257 **3.1 Changes in PDSI and drought characteristics**

258 We present the changes in multi-model ensemble mean PDSI from the baseline period

259 (1986-2005) to each of the 1.5$^o$C and 2$^o$C scenarios and model consistency in Figure 3. For the

260 1.5$^o$C warmer world, the PDSI would decrease (more drought-prone) with relatively higher model

261 consistency (6~11 models in totally 11 climate models) in some regions, for example, Amazon

262 (0.7±0.8 -> -0.1±0.2), Northeastern Brazil (0.5±0.6 -> -0.1±0.3), Southern Europe and

263 Mediterranean (0.4±0.6 -> -0.3±0.2), Central America and Mexico (0.2±0.4 -> -0.2±0.1), Central

264 Europe (0.3±1.0 -> -0.1±0.4) as well as Southern Africa (0.5±0.5 -> -0.3±0.2); slightly increase (less

265 drought-prone) in Alaska/Northwest Canada (-0.01±0.5 -> -0.3±0.2) and North Asia (-0.1±1.0 ->

266 -0.2±0.2) but with relatively low model consistency. The geographic pattern of changes in PDSI for

the 2°C warmer world is quite similar to that of 1.5°C warmer world, but the magnitude of
change would intensify (in both direction) East Canada, Greenland, Iceland (-0.3±0.2-> -0.4±0.2),
East Africa (-0.5±0.2-> -0.3±0.2), Northern Europe (-0.3±0.3-> -0.2±0.3), East Asia (-0.3±0.1->
-0.2±0.4), South Asia (-1.0±1.2-> -0.8±0.3) and West Africa (-0.3±0.2-> -0.3±0.3). When global
warming is capped at 1.5°C instead of 2°C above the pre-industrial levels, the PDSI value would
elevate at the globe (66°S-66°S, -0.4±0.2-> -0.3±0.2) and most regions (Alaska/Northwest Canada,
East Africa, West Africa, Tibetan Plateau, North Asia, East Asia, South Asia and Southeast Asia)
(Figure 4).

<Figure 3, here, thanks>

<Figure 4, here, thanks>

We analyze the changes in drought characteristics such as its duration, severity and intensity in
1.5°C and 2°C warmer worlds. In terms of the drought duration (Figures 5-6), we find robust
large-scale features. For example, the drought duration would generally increase at the globe
(2.9±0.5 -> 3.1±0.4 months and 2.9±0.5 -> 3.2±0.5 months from the baseline period to the 1.5°C
and 2°C scenarios) and most regions (especially for Amazon, Sahara, Northeastern Brazil and
North Australia) except for North Asia (2.7±0.6 -> 2.6±0.5 months and 2.7±0.6 -> 2.5±0.4 months)
in both worlds. The high model consistency in most regions (i.e., Amazon, Sahara and
Northeastern Brazil) for both warming targets gives us more confidence on these projections.
Relative to the 2°C warmer target, a 1.5°C warming target is more likely to reduce drought
duration at both global and regional scales (except for Alaska/Northwest Canada, East Africa,
Sahara, North Europe, North Asia, South Asia, Southeast Asia, Tibetan Plateau and West Africa).

<Figure 5, here, thanks>

\<Figure 6, here, thanks\>

Drought intensity and drought severity are commonly used for quantifying the extent of water
availability drops significantly below normal conditions in a region. In this study, the drought
intensity is projected to increase at the globe ($0.9\pm0.3$ -> $1.1\pm0.3$ and $0.9\pm0.3$ -> $1.0\pm0.2$ from the
baseline period to the $1.5^{o}$C and $2^{o}$C scenarios) and in most of the regions except for North Asia,
Southeast Asia and West Africa in $1.5^{o}$C and $2^{o}$C warmer worlds (Figures 7-8). Compare to the $2^{o}$C
warmer world, the drought intensity would obviously relieve at the global and sub- continental
scales except for East Canada, Greenland, Iceland ($1.0\pm0.6$ -> $0.8\pm0.5$) and West North America
($0.9\pm0.3$ -> $0.8\pm0.2$) in the $1.5^{o}$C warmer world. In addition, the projected drought severity would
also increase in these warmer worlds at the globe ($3.0\pm1.9$ -> $4.5\pm3.0$ and $3.0\pm1.9$ -> $3.8\pm2.0$ from
the baseline period to the $1.5^{o}$C and $2^{o}$C warmer worlds) and in most regions except for North
Asia ($1.8\pm0.6$ -> $1.8\pm0.7$ and $1.8\pm0.6$ -> $1.5\pm0.3$) (Figures 9-10). When global warming is
maintained at $1.5^{o}$C instead of $2^{o}$C above the pre-industrial levels, the drought severity would
weaken in most regions except for Sahara ($3.1\pm0.9$-> $3.5\pm1.3$), North Asia ($1.5\pm0.3$ -> $1.8\pm0.8$),
Southeast Asia ($17.2\pm20.1$ -> $35.8\pm57.2$), and West North America ($2.4\pm1.7$ -> $2.5\pm1.4$). The
projected uncertainties are relatively low (6~11 models in totally 11 models) for the changes of
each drought characteristic in these warming scenarios all over the world, except for some parts
of Alaska/Northwest Canada, East Canada, Greenland, Iceland, West North America, Central
North America, East North America, Sahara, West Africa, East Africa and North Asia.

\<Figure 7, here, thanks\>

\<Figure 8, here, thanks\>

**3.2 Impact of severe drought on population**
To understand the societal influences of severe drought, we combine the drought projection with
SSP1 population information and estimate the total, urban and rural population affected by
severe drought in the baseline period, 1.5$^o$C and 2$^o$C warmer worlds (Figures 11-13). Compare to
the baseline period, the frequency of severe drought (PDSI < -3), drought-affected total and
urban population would increase in most of the regions in 1.5$^o$C and 2$^o$C warmer worlds. Globally,
we estimate that 132.5±216.2 million (350.2±158.8 million urban population and -217.7±79.2
million rural population) and 194.5±276.5 million (410.7±213.5 million urban population and
-216.2±82.4 million rural population) additional people would be exposed solely to severe
droughts in the 1.5$^o$C and 2$^o$C warmer worlds, respectively. The severe drought affected total
population would increase under these warming targets in most regions, except for East Asia,
North Asia, South Asia, Southeast Asia, Tibetan Plateau and West Coast South America.

<Figure 9, here, thanks>

<Figure 10, here, thanks>

The severe drought affect urban (rural) population would increase (decrease) in all global regions
in 1.5$^o$C and 2$^o$C warmer worlds. For example, the projections suggest that more urban
population would be exposed to severe drought in Central Europe (10.9±7.7 million), Southern
Europe and Mediterranean (14.0±4.6 million), West Africa (65.3±34.1 million), East Asia
(16.1±16.0 million), West Asia (16.2±7.4 million) and Southeast Asia (24.4±19.7 million) in 1.5$^o$C
warmer world relative to the baseline period. We also find that the number of affected people
would escalate further in these regions in 2$^o$C warmer world. In terms of the rural population,
less people in Central Asia (-4.1±4.7 million and -3.3±4.1 million for the 1.5$^o$C and 2$^o$C warmer
worlds), Central North America (-0.5±1.1 million and -0.4±0.9 million), Southern Europe and
Mediterranean (-3.6±3.2 million and -2.9±3.8 million), South Africa (-3.3±1.5 million and -2.9±1.8
million), Sahara (-1.0±2.5 million and -0.9±2.9 million), South Asia (-70.2±29.7 million and
-72.9±30.0 million), Tibetan Plateau (-2.3±1.8million and -2.1±1.9 million) and West North
America (-1.7±1.0 million and -1.6±1.1 million) would be exposed to the severe drought in
$1.5^{o}$C and $2^{o}$C warmer worlds relative to the baseline period. The distinct influences of severe
drought on urban and rural population are driven by both climate warming and population
growth, especially by the urbanization-induced population migration.

<Figure 11, here, thanks>

<Figure 12, here, thanks>

When global warming approaches $1.5^{o}$C (instead of $2^{o}$C) above the pre-industrial levels, relatively
less total, urban and rural population (except for East Africa and South Asia) would be affected
despite more frequent severe drought in most regions such as East Asia, Southern Europe and
Mediterranean, Central Europe and Amazon. This implies that the benefit of holding global
warming at $1.5^{o}$C instead of $2^{o}$C is apparent to the severe-drought affected total, urban and rural
population in most regions, but challenges remain in the East Africa and South Asia.

<Figure 13, here, thanks>

**4 Discussions**
The changes in PDSI, drought duration, intensity and severity with climate warming (i.e., $1.5^{o}$C
and $2^{o}$C warmer worlds) projected in this study is in general agreement with that concluded by
IPCC (2013) despite regional variation. For example, as revealed in this study, the gradual decline
of PDSI (drought-prone) in American Southwest and Central Plains was also projected using an
empirical drought reconstruction and soil moisture metrics from 17 state–of-the-art GCMs in the
21$^{st}$ century (Cook et al., 2015). The ascending risk of drought in Sahara, North Australia and
South Africa coincided with Huang et al. (2017), which projected that global drylands would
degrade in 2$^{o}$C warmer world. Moreover, the increases in drought duration, intensity and severity
in Central America, Amazon, South Africa and Mediterranean are in agreement with the
extension of dry spell length and less water availability in these regions under the 1.5$^{o}$C and/or
2$^{o}$C warming scenarios (Schleussner et al., 2016; Lehner et al., 2017). In addition, we find that the
affected population attributes more (50%~75%) to the population growth rather than the climate
change driven severe drought in the 1.5$^{o}$C and 2$^{o}$C warmer worlds. This number is greater than
that concluded by Smirnov et al. (2016), maybe due to different study periods, population data,
drought index and warming scenarios used.

Projections presented in current study inherited several sources of uncertainty. Firstly, there are
considerable uncertainties in the numerical projections from different climate models under
varied greenhouse gas emission scenarios, especially on a regional scale (i.e., Sahara,
Alaska/Northwest Canada and North Asia). However, the utility of multiple GCMs and emission
scenarios should allow us to synthesize future projections better than single model/scenario
analysis (Schleussner et al., 2016; Wang et al., 2017; Lehner et al., 2017). On top of that, we
performed uncertainty analysis such as understanding the model consistency (e.g.,
increase/decrease) and inter-model variance (for magnitude changes). These enable us to
characterize regional and global projections which could vary due to different model structure of
GCMs and how they behave under different RCP scenarios. Moreover, the global and regional
responses (i.e., warming/precipitation patterns) to varied warming scenarios (i.e., 1.5$^{o}$C and 2$^{o}$C)
showed little dependences on RCP scenarios (King et al., 2017; Hu et al., 2017). Therefore, the
uncertainty caused by the choice of RCP scenarios might be small. Secondly, there are various
ways of picking the 1.5$^o$C or 2$^o$C warming signals (King et al., 2017). Current study considered
both the influences of multi-model and multi-scenario for each warming scenario using the
20-year smoothed multi-model ensemble mean GMT. The selected periods of 1.5$^o$C and 2.0$^o$C
warmer worlds are close to that of King et al. (2017). Finally, the SSP1 population data and the
single drought index used might introduce uncertainties. Despite these sources of uncertainty,
these projections are quite robust with high model consistency across most regions.

This analysis evaluated the risk of droughts in terms of how they would change in the future
period (1.5$^o$C or 2$^o$C warmer worlds) relative to the baseline period; and the difference between
the two warmer worlds. In this perspective, uncertainty arises from climate model bias in
between two periods more or less cancels each other. Previous studies demonstrated that
bias-corrections do not yield much difference in such circumstance (e.g., Sun et al., 2011; Maraun,
2016). In addition, the methodology here requires the meteorological information with physical
meaning (see Section 2.1) that is consistent with the energy balance of the climate model
(Equation 3 in Section 2.3), hence existing bias-correction measures (with known weakness in
maintaining the physical aspect of bias-corrected output) appears less feasible. (Future
innovation which accounts for both statistics and energy balance of climate model output in new
bias-correction methodology for handling the high non-linear outcomes should be a subject of
scientific interest.) The rationale of using model consistency (Figures 3, 5, 7, 9) as a form of
"confidence index" here emerges from the idea that, whilst model validation in historical period
is helpful, it does not necessary reveal the ability of each climate model in projection of risk
change. Thus this kind of confidence index is informative for synthesizing multi-model projections,
probably explains why it is still common in many global studies involving multi-model ensembles
(e.g., Hirabayashi et al., 2013; Koirala et al., 2014).

**5 Conclusions**
Motivated by the 2015 Paris Agreement proposal, we analyzed the CMIP5 GCM output and
presented the first comprehensive assessment of changes in drought characteristics and the
potential impacts of severe drought on population (total, urban and rural) in $1.5^{o}$C and $2^{o}$C
warmer worlds. We found that the risk of drought would increase (i.e., decrease in PDSI, increase
in drought duration, drought intensity and drought severity) globally and most regions (i.e.,
Amazon, Northeastern Brazil, Central Europe) for in $1.5^{o}$C and $2^{o}$C warmer worlds relative to the
baseline period (1986-2005). However, the amplitudes of change in drought characteristics vary
among the regions. Relative to the $2^{o}$C warming target, a $1.5^{o}$C warming target is more likely to
reduce drought risk (less drought duration, drought intensity and drought severity but relatively
more frequent severe drought) significantly on both global and regional scales. The high model
consistency (6~11 out of 11 GCMs) across most regions (especially Amazon, Sahara and
Northeastern Brazil) gives us more confidence on these projections.

Despite the uncertainties inherited from the GCMs and population data used, as well as the
definition of the $1.5^{o}$C and $2^{o}$C periods, we found significant changes of drought characteristics
under both warming scenarios and societal impacts of severe drought by limiting temperature
target at 1.5 $^{o}$C instead of 2 $^{o}$C in several hotspot regions. More total (+132.5±216.2 million and
+194.5±276.5 million globally) and urban population (+350.2±158.8 million and +410.7±213.5
million globally) would be exposed to severe drought in most regions (especially East Africa, West
Africa and South Asia) in 1.5 $^{o}$C and 2 $^{o}$C warmer worlds, particularly for the latter case.

Meanwhile, less rural population (-217.7±79.2 million and -216.2±82.4 million globally) in, e.g.,
Central Asia, East Canada, Greenland, Iceland, Central North America, Southern Europe and
Mediterranean, North Australia, South Africa, Sahara, South Asia, Tibetan Plateau and West
North America would be affected. When global mean temperature increased by 1.5 $^{o}$C instead of
2 $^{o}$C above the pre-industrial level, the total, urban and rural population affected by severe
drought would decline in most regions except for East Africa and South Asia.

In general, this comprehensive global drought risk assessment should provide useful insights for
international decision-makers to develop informed climate policy within the framework of the
2015 Paris Agreement. Whist most regions would benefit from reduced societal impacts in the
1.5 $^{o}$C wamer world, local governments in East Africa and South Asia should be prepared to deal
with drought-driven challenges (see paragraph above). Future studies on understanding the
causes of changes in global and regional droughts (e.g., changing pattern/duration of
precipitation and evaporative demand) with respect to these warming targets should assist
drought risk adaptation and mitigation planning.

**Data availability.** The datasets applied in this study are available at the following locations:
— CMIP5 model experiments (Taylor et al., 2012), https://esgf-data.dkrz.de/projects/

esgf-dkrz/

— Spatial population scenarios (Shared Socioeconomic Pathway 1, SSP1, Jones and O'Niell,

2016), https://www2.cgd.ucar.edu/sections/tss/iam/spatial-population-scenarios

**Competing interests.** The authors declare that they have no conflict of interest.
**Acknowledgements.** This study was financially supported by the National Research and
Development Program of China (2016YFA0602402 and 2016YFC0401401), the National Natural
Sciences Foundation of China (41401037), the Key Research Program of the Chinese Academy of
Sciences (ZDRW-ZS-2017-3-1), the CAS Pioneer Hundred Talents Program (Fubao Sun) and the
CAS President's International Fellowship Initiative (Wee Ho Lim, 2017PC0068). We thank the
Editor (Michel Crucifix), Dimitri Defrance and an anonymous reviewer for their helpful comments.

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

**Table 1:** Details of CMIP5 climate models applied in this study

| Climate models | abbreviation | Horizontal Resolution | Future Scenarios |
|---|---|---|---|
| ACCESS1.0 | ACCESS | 1.300×1.900 degree | RCP4.5, RCP8.5 |
| BCC_CSM1.1 | BCC | 2.813×2.791 degree | RCP4.5, RCP8.5 |
| BNU-ESM | BNU | 2.810×2.810 degree | RCP4.5, RCP8.5 |
| CanESM23 | CANESM | 2.813×2.791 degree | RCP4.5, RCP8.5 |
| CNRM-CM5 | CNRM | 1.406×1.401 degree | RCP4.5, RCP8.5 |
| CSIRO Mk3.6.0 | CSIRO | 1.875×1.866 degree | RCP4.5, RCP8.5 |
| GFDL CM3 | GFDL | 2.500×2.000 degree | RCP4.5, RCP8.5 |
| INM-CM4.0 | INM | 2.000×1.500 degree | RCP4.5, RCP8.5 |
| IPSL-CM5B-LR | IPSL | 1.875×3.750 degree | RCP4.5, RCP8.5 |
| MRI-CGCM3 | MRI | 1.125×1.125 degree | RCP4.5, RCP8.5 |
| MIROC-ESM | MIROC | 2.813×2.791 degree | RCP4.5, RCP8.5 |

**Table 2:** Definition of regions in this study, after IPCC (2012)


| ID | abbreviation | Regional Representation |
|----|--------------|------------------------|
| 1 | ALA | Alaska/Northwest Canada |
| 2 | CGI | East Canada, Greenland, Iceland |
| 3 | WNA | West North America |
| 4 | CNA | Central North America |
| 5 | ENA | East North America |
| 6 | CAM | Central America and Mexico |
| 7 | AMZ | Amazon |
| 8 | NEB | Northeastern Brazil |
| 9 | WSA | West Coast South America |
| 10 | SSA | Southeastern South America |
| 11 | NEU | Northern Europe |
| 12 | CEU | Central Europe |
| 13 | MED | Southern Europe and Mediterranean |
| 14 | SAH | Sahara |
| 15 | WAF | West Africa |
| 16 | EAF | East Africa |
| 17 | SAF | Southern Africa |
| 18 | NAS | North Asia |
| 19 | WAS | West Asia |
| 20 | CAS | Central Asia |
| 21 | TIB | Tibetan Plateau |
| 22 | EAS | East Asia |
| 23 | SAS | South Asia |
| 24 | SEA | Southeast Asia |
| 25 | NAU | North Australia |
| 26 | SAU | South Australia/New Zealand |
| 27 | GLOBE | Globe |

**Figure captions:**

**Figure 1:** Definition of the baseline period, 1.5$^o$C and 2$^o$C warmer worlds based on CMIP5 GCM-simulated changes in global mean temperature (GMT, relative to the pre-industrial levels: 1850-1900). The dark blue and dark yellow shadows indicate the 25th and 75th percentiles of multi-model simulated GMT for RCP 4.5 and RCP 8.5 scenarios, respectively. Both the multi-model ensemble mean and percentiles shown in the figure are smoothed using a moving average approach in a 20-year window

**Figure 2:** Palmer Drought Severity Index (PDSI)-based drought characteristics definition through the run theory

**Figure 3:** Changes in multi-model ensemble mean PDSI (i) and model consistency (ii) on a spatial resolution of 0.5$^o$ × 0.5$^o$, (a) from the baseline period to 1.5$^o$C warmer world, (b) from the baseline period to 2$^o$C warmer world and (c): (a)-(b). Robustness of projections increases with higher model consistency and vice-versa. The dark-gray boxes show the world regions adopted by IPCC (2012), which are labeled in (a)(i) using the ID numbers defined in Table 2. Legend in (a)(i) applies to (b)(i) and (c)(i); legend in (a)(ii) applies to (b)(ii) and (c)(ii).

**Figure 4:** Multi-model projected PDSI at the globe (66$^o$N-66$^o$S) and in 27 world regions for the baseline period, 1.5$^o$C and 2$^o$C warmer worlds. The projected uncertainty of multiple climate models is shown through box plots for each region and for each period

**Figure 5:** Changes in multi-model ensemble mean drought duration (months) (i) and model consistency (ii) on a spatial resolution of 0.5$^o$ × 0.5$^o$, (a) from the baseline period to 1.5$^o$C warmer world, (b) from the baseline period to 2$^o$C warmer world and (c) : (a)-(b). The dark-gray boxes show the regions adopted by IPCC (2012), which are labeled in (a)(i) using the ID numbers defined in Table 2. Legend in (a)(i) applies to (b)(i) and (c)(i); legend in (a)(ii) applies to (b)(ii) and (c)(ii).

**Figure 6:** Multi-model projected drought duration (months) at the globe (66$^o$N-66$^o$S) and in 27 world regions for the baseline period, 1.5$^o$C and 2$^o$C warmer worlds. The projected uncertainty of multiple climate models is shown through box plots for each region and for each period

**Figure 7:** Changes in multi-model ensemble mean drought intensity (dimensionless) (i) and model consistency (ii) on a spatial resolution of 0.5$^o$ × 0.5$^o$, (a) from the baseline period to 1.5$^o$C warmer world, (b) from the baseline period to 2$^o$C warmer world and (c): (a)-(b). The dark-gray boxes show the regions adopted by IPCC (2012), which are labeled in (a)(i) using the ID numbers defined in Table 2. Legend in (a)(i) applies to (b)(i) and (c)(i); legend in (a)(ii) applies to (b)(ii) and (c)(ii).

**Figure 8:** Multi-model projected drought intensity (dimensionless) at the globe (66$^o$N-66$^o$S) and in 27 world regions for the baseline period, 1.5$^o$C and 2$^o$C warmer worlds. The projected uncertainty of multiple climate models is shown through box plots for each region and for each period

**Figure 9:** Changes in multi-model ensemble mean drought severity (dimensionless) (i) and model consistency (ii) on a spatial resolution of 0.5$^o$ × 0.5$^o$, (a) from the baseline period to 1.5$^o$C warmer world, (b) from the baseline period to 2$^o$C warmer world and (c): (a)-(b). The dark-gray boxes show the regions adopted by IPCC (2012), which are labeled in (a)(i) using the ID numbers defined in Table 2. Legend in (a)(i) applies to (b)(i) and (c)(i); legend in (a)(ii) applies to (b)(ii) and (c)(ii).

**Figure 10:** Multi-model projected drought severity (dimensionless) at the globe (66$^o$N-66$^o$S) and in 27 world regions for the baseline period, 1.5$^o$C and 2$^o$C warmer worlds. The projected uncertainty of multiple climate models is shown through box plots for each region and for each

period
**Figure 11:** Multi-model projected frequency (Freq.) and affected total population (Pop., million)
of severe drought (PDSI < -3) at the globe and in 27 world regions for the baseline period (black,
fixed SSP1 2000 population), 1.5$^\circ$C (orange, fixed SSP1 2100 population) and 2$^\circ$C (red, fixed SSP1
2100 population) warmer worlds. The projected uncertainties (standard deviation of
multiple-model results) of multiple climate models are shown by error bars (horizontal and
vertical)
**Figure 12:** Multi-model projected frequency (Freq.) and affected urban population (Pop., million)
of severe drought (PDSI < -3) at the globe and in 27 regions for the baseline period (black, fixed
SSP1 2000 population), 1.5 $^\circ$C (orange, fixed SSP1 2100 population) and 2 $^\circ$C (red, fixed SSP1 2100
population) warmer worlds. The projected uncertainties (standard deviation of multiple-model
results) of multiple climate models are shown by error bars (horizontal and vertical)
**Figure 13:** Multi-model projected frequency (Freq.) and affected rural population (Pop., million)
of severe drought (PDSI < -3) at the globe and in 27 regions for the baseline period (black, fixed
SSP1 2000 population), 1.5 $^\circ$C (orange, fixed SSP1 2100 population) and 2 $^\circ$C (red, fixed SSP1 2100
population) warmer worlds. The projected uncertainties (standard deviation of multiple-model
results) of multiple climate models are shown by error bars (horizontal and vertical)

**Figure 1:** Definition of the baseline period, 1.5$^{\circ}$C and 2$^{\circ}$C warmer worlds based on CMIP5 GCM-simulated changes in global mean temperature (GMT, relative to the pre-industrial levels: 1850-1900). The dark blue and dark yellow shadows indicate the 25[th] and 75[th] percentiles of multi-model simulated GMT for RCP 4.5 and RCP 8.5 scenarios, respectively. Both the multi-model ensemble mean and percentiles shown in the figure are smoothed using a moving average approach in a 20-year window

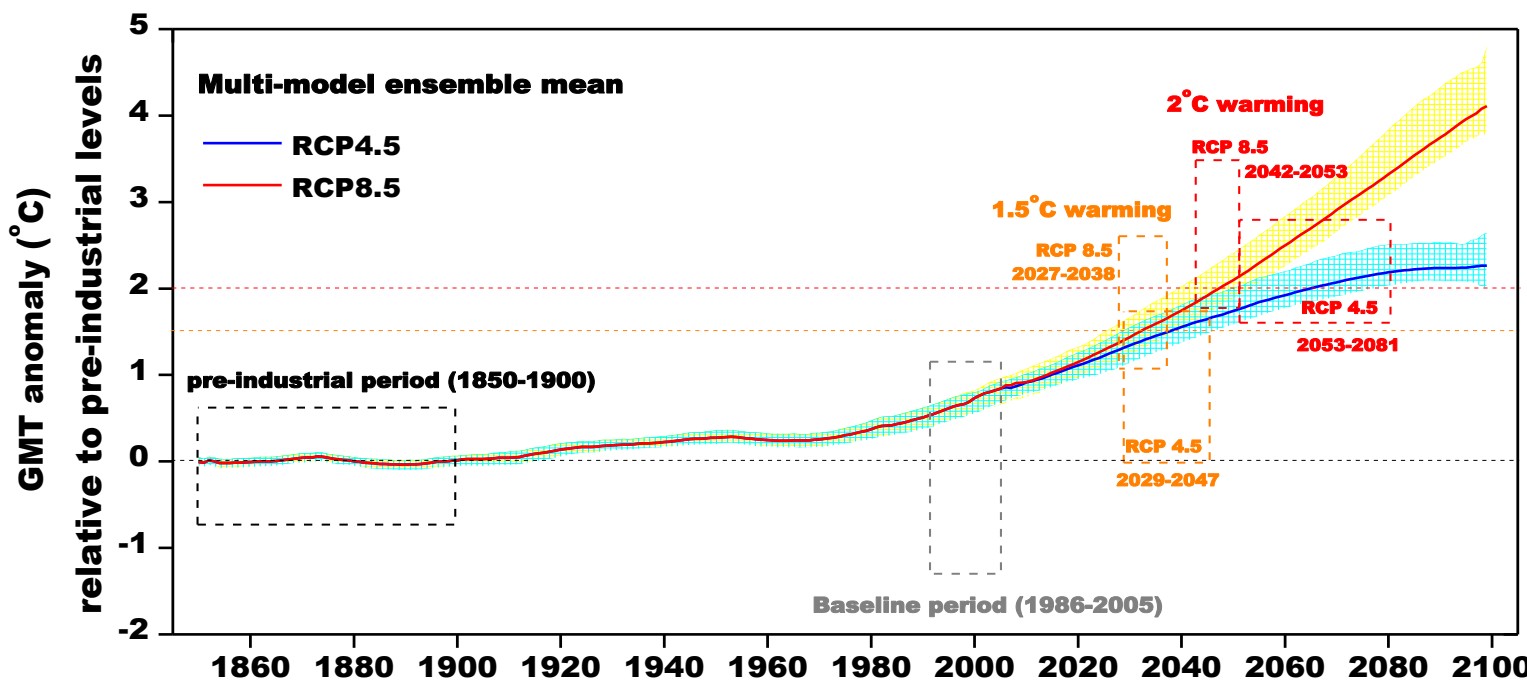

**Figure 2:** Palmer Drought Severity Index (PDSI)-based drought characteristics definition through the run theory

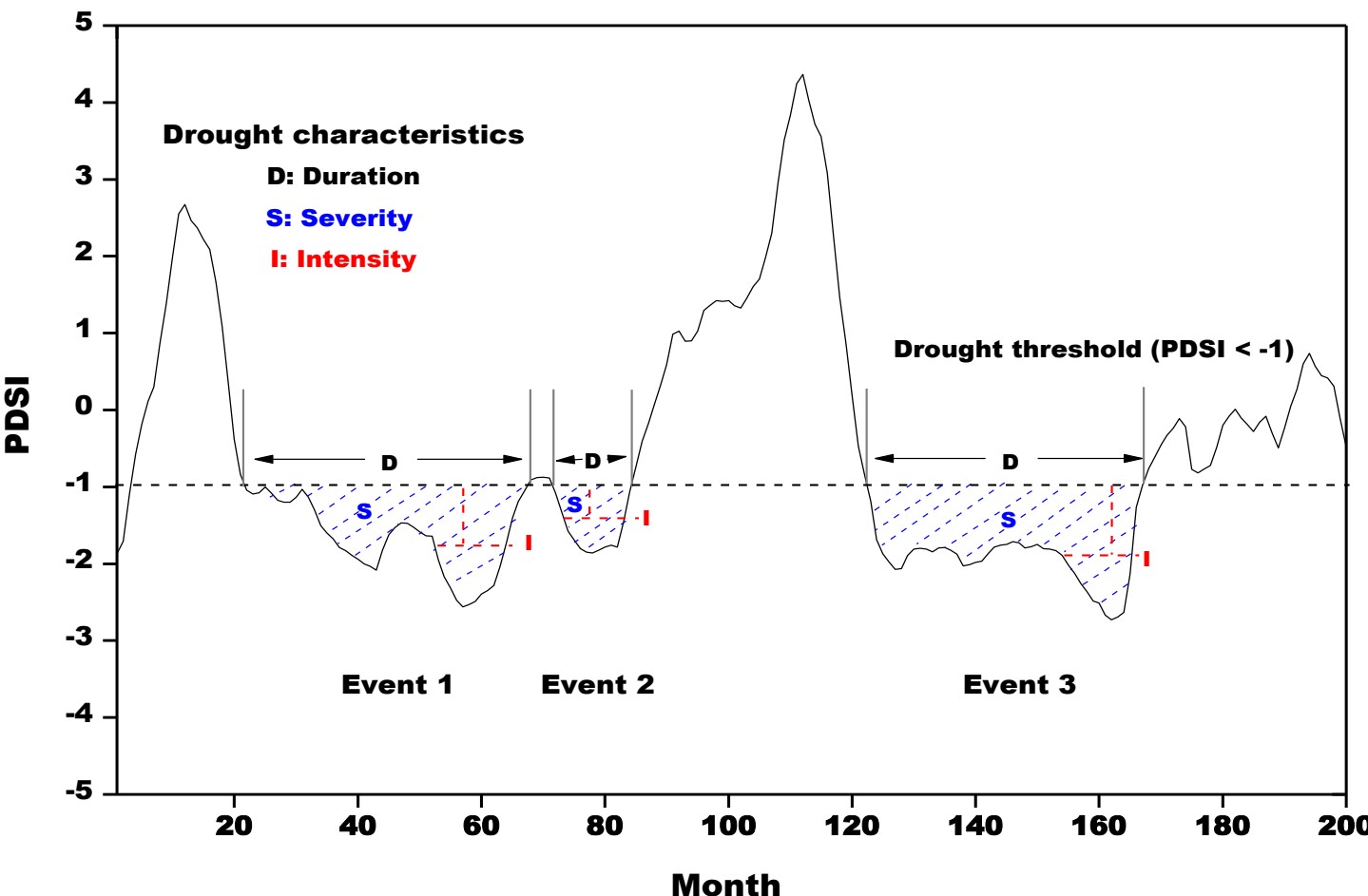


**Figure 3:** Changes in multi-model ensemble mean PDSI (i) and model consistency (ii) on a spatial resolution of $0.5^{o} \times 0.5^{o}$, (a) from the baseline
period to 1.5$^{o}$C warmer world, (b) from the baseline period to 2$^{o}$C warmer world and (c): (a)-(b). Robustness of projections increases with
higher model consistency and vice-versa. The dark-gray boxes show the world regions adopted by IPCC (2012), which are labeled in (a)(i) using
the ID numbers defined in Table 2. Legend in (a)(i) applies to (b)(i) and (c)(i); legend in (a)(ii) applies to (b)(ii) and (c)(ii).

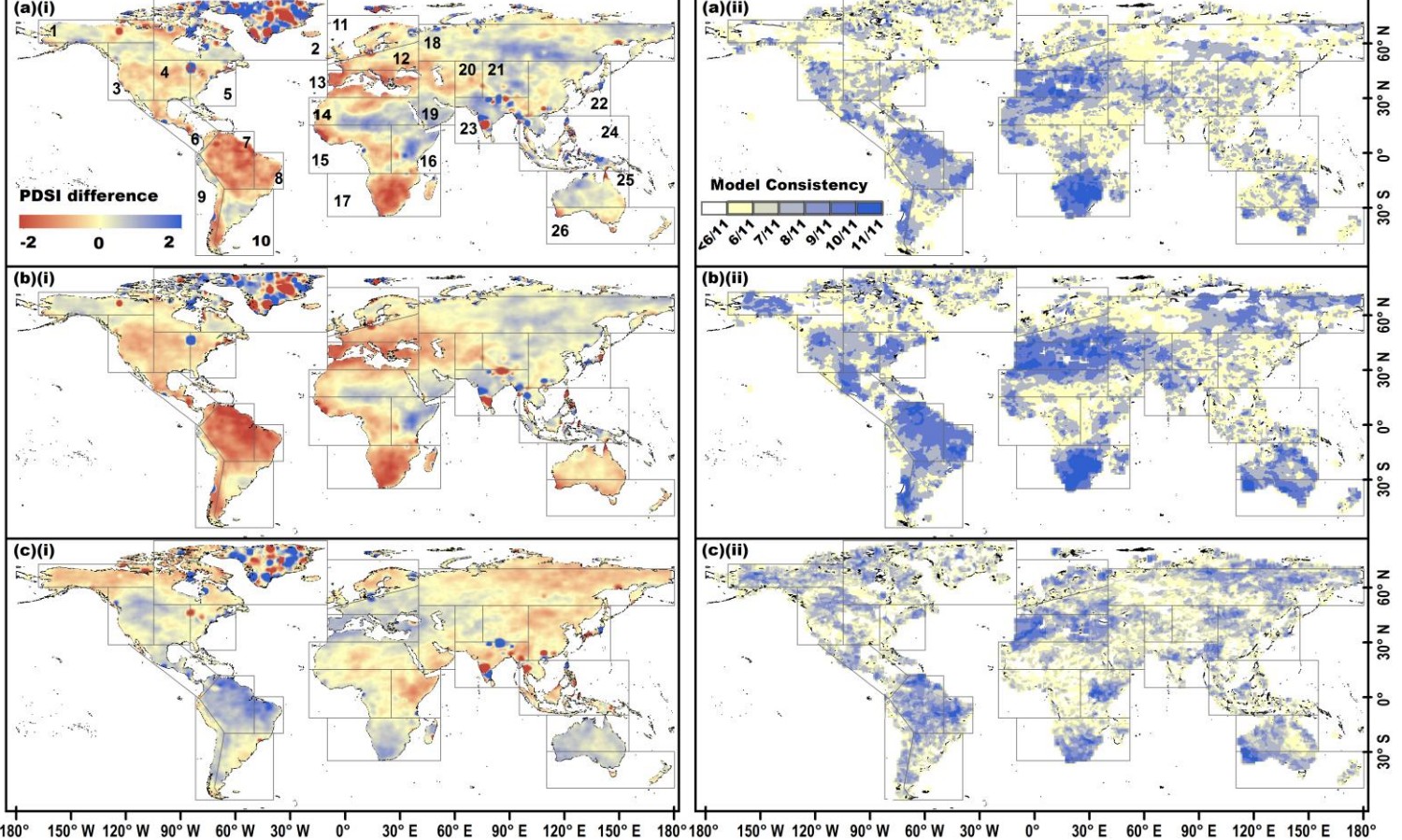


**Figure 4:** Multi-model projected PDSI at the globe (66°N-66°S) and in 27 world regions for the baseline period, 1.5°C and 2°C warmer worlds.
The projected uncertainty of multiple climate models is shown through box plots for each region and for each period

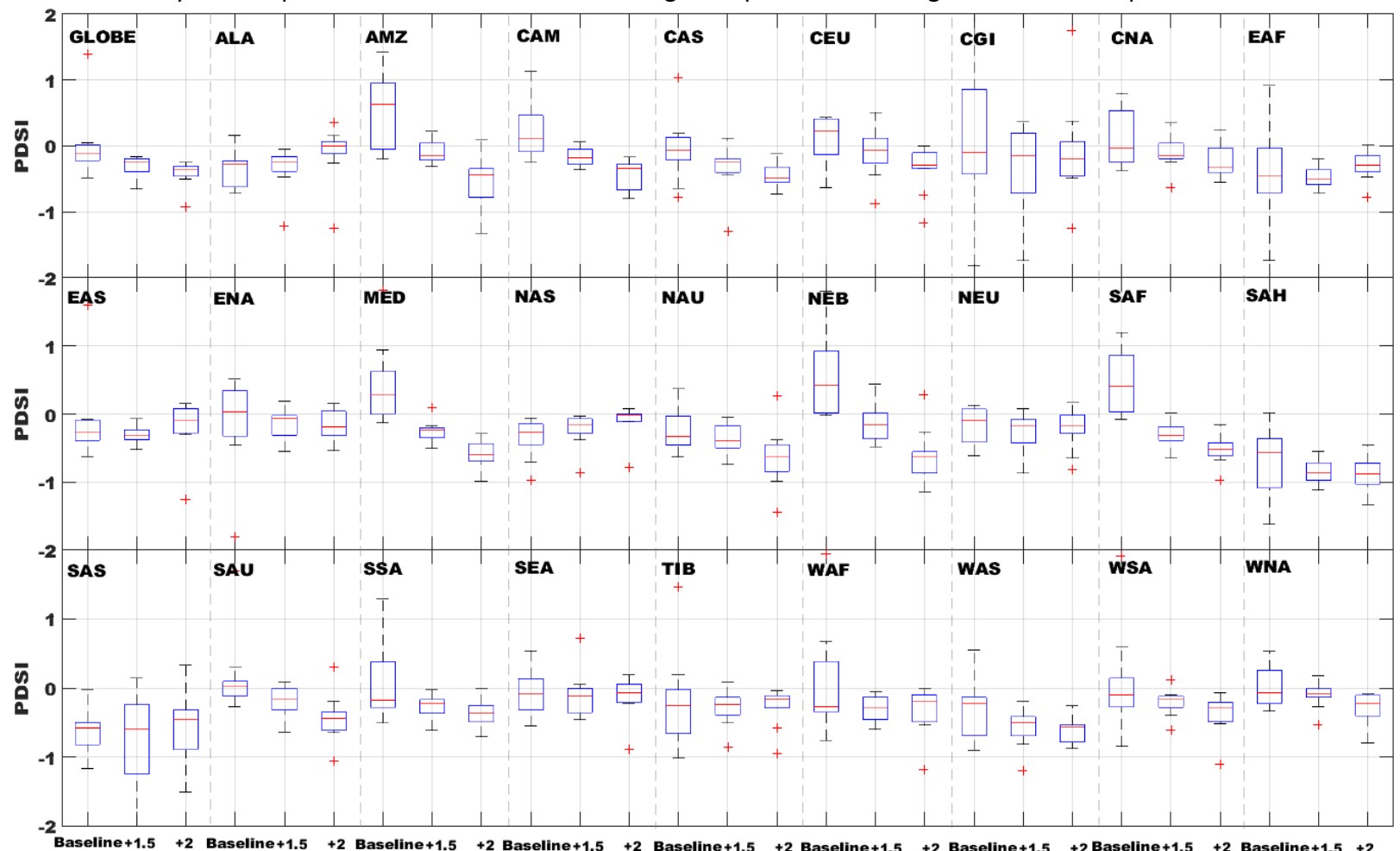


**Figure 5:** Changes in multi-model ensemble mean drought duration (months) (i) and model consistency (ii) on a spatial resolution of 0.5$^{o}$ × 0.5$^{o}$,
(a) from the baseline period to 1.5$^{o}$C warmer world, (b) from the baseline period to 2$^{o}$C warmer world and (c): (a)-(b). The dark-gray boxes
show the regions adopted by IPCC (2012), which are labeled in (a)(i) using the ID numbers defined in Table 2. Legend in (a)(i) applies to (b)(i)
and (c)(i); legend in (a)(ii) applies to (b)(ii) and (c)(ii).

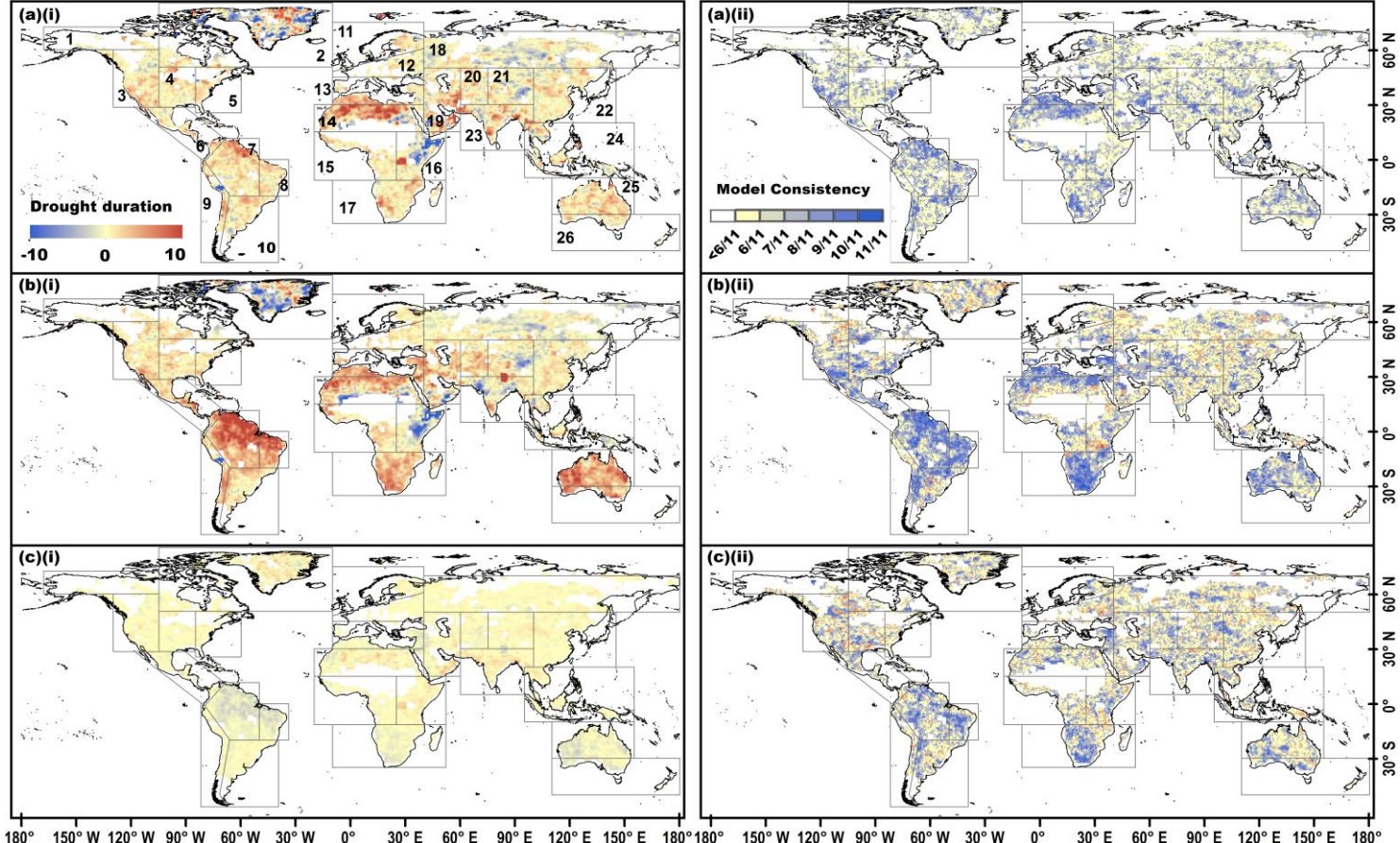


**Figure 6:** Multi-model projected drought duration (months) at the globe (66°N-66°S) and in 27 regions for the baseline period, 1.5°C and 2°C
warmer worlds. The projected uncertainty of multiple climate models is shown through box plots for each region and for each period

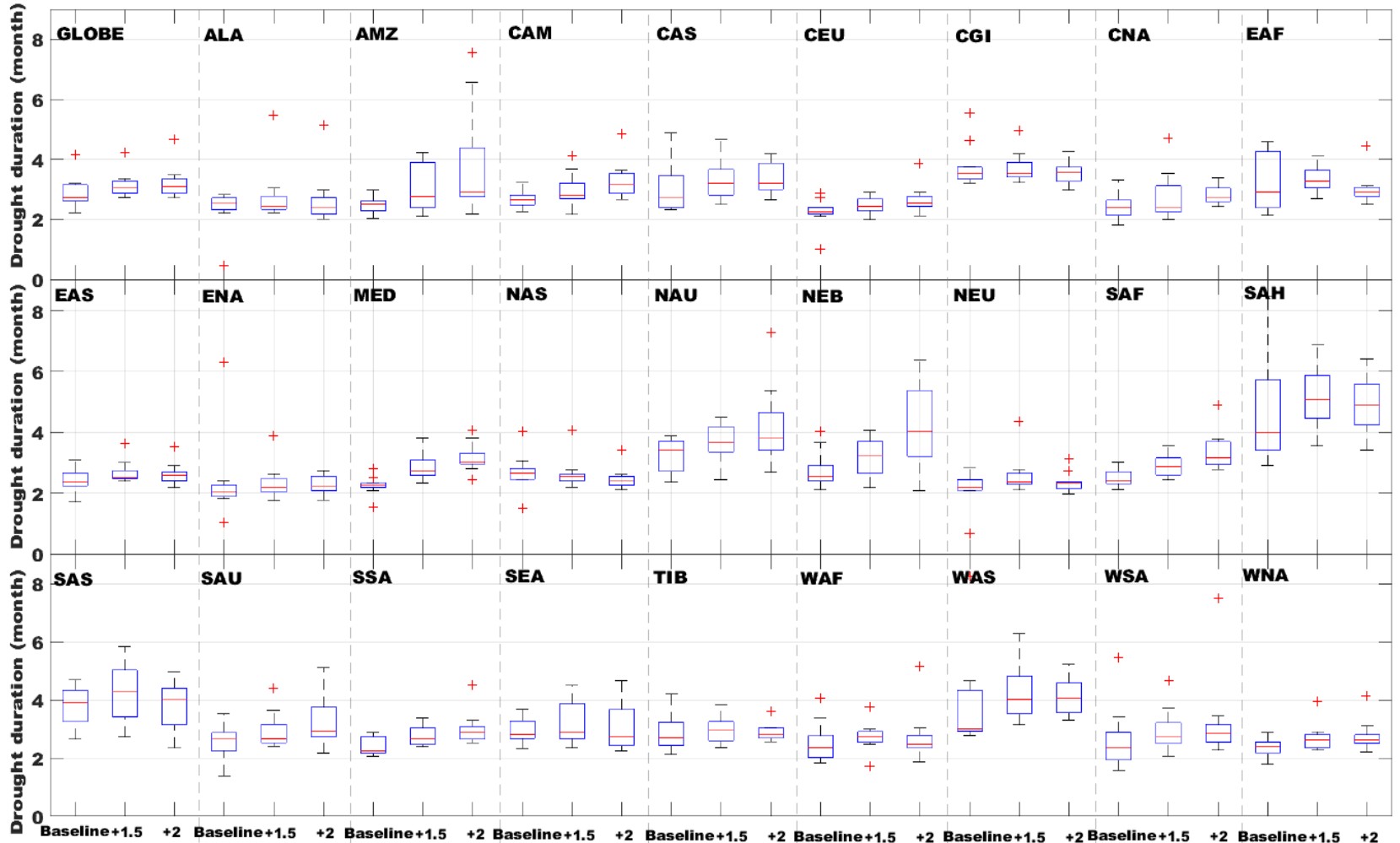


**Figure 7:** Changes in multi-model ensemble mean drought intensity (dimensionless) (i) and model consistency (ii) on a spatial resolution of 0.5$^{o}$
× 0.5$^{o}$, (a) from the baseline period to 1.5$^{o}$C warmer world, (b) from the baseline period to 2$^{o}$C warmer world and (c): (a)-(b). The dark-gray
boxes show the regions adopted by IPCC (2012), which are labeled in (a)(i) using the ID numbers defined in Table 2. Legend in (a)(i) applies to
(b)(i) and (c)(i); legend in (a)(ii) applies to (b)(ii) and (c)(ii).

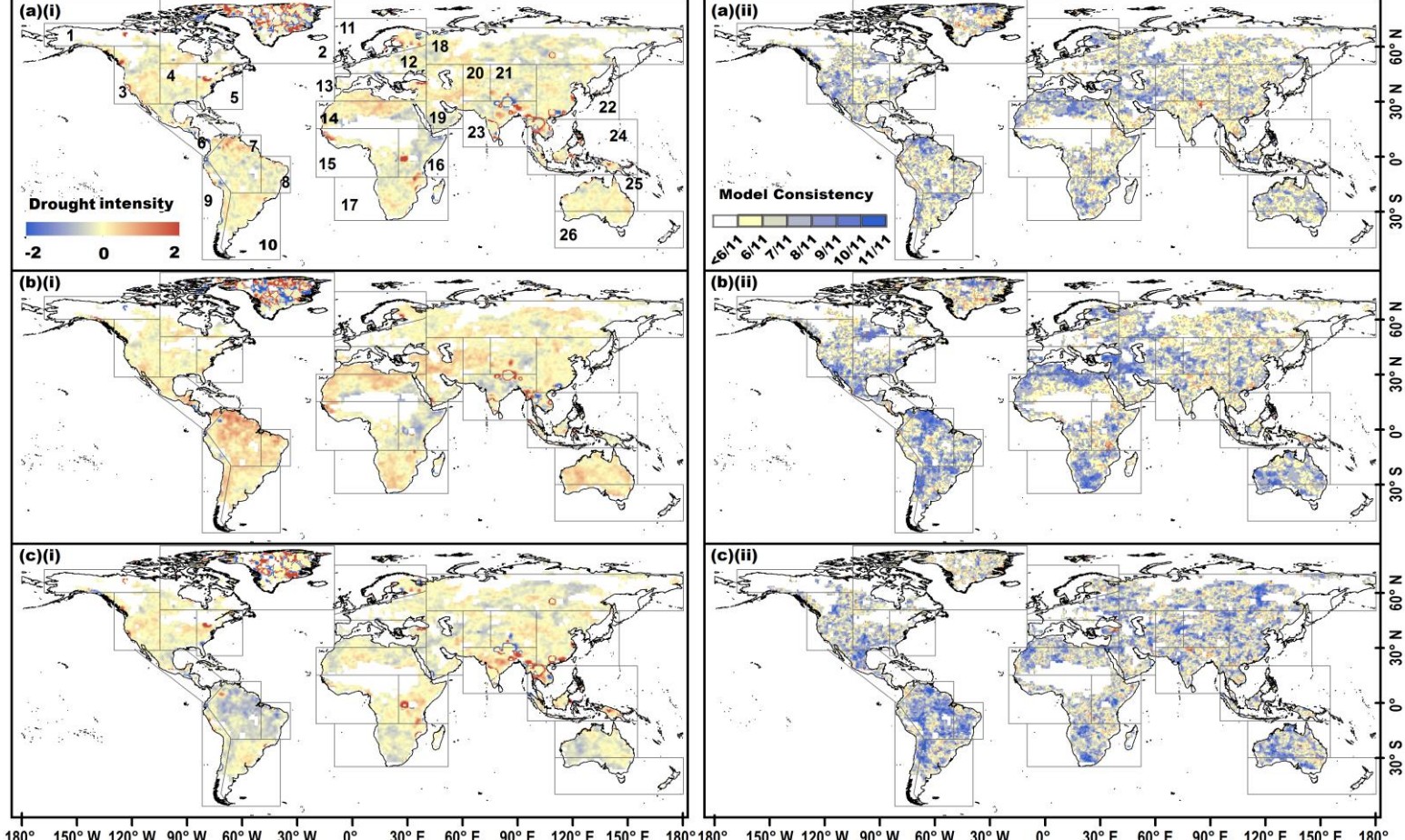


 **Figure 8:** Multi-model projected drought intensity (dimensionless) at the globe (66°N-66°S) and in 27 regions for the baseline period, 1.5°C and
 2°C warmer worlds. The projected uncertainty of multiple climate models is shown through box plots for each region and for each period

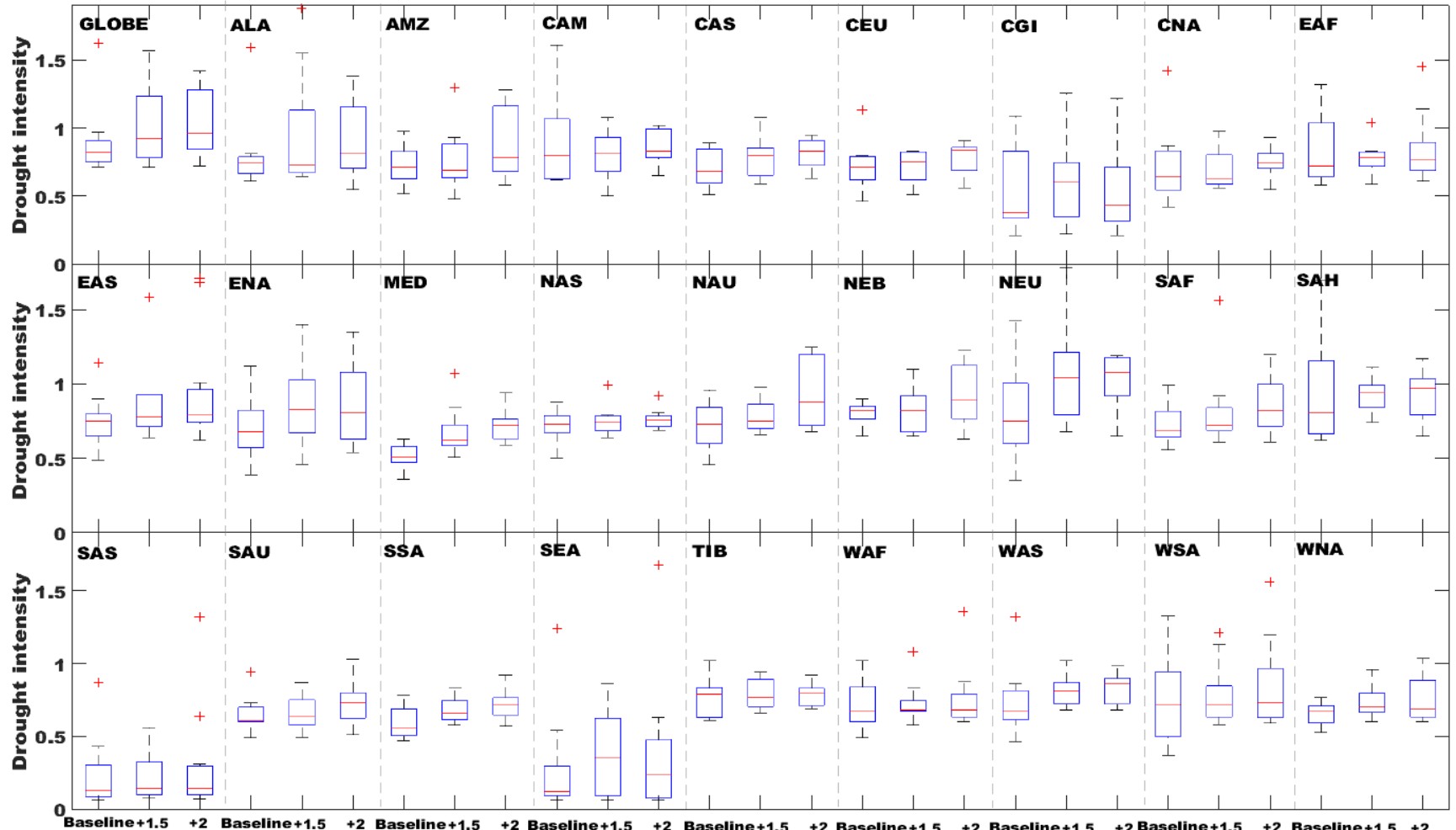

**Figure 9:** Changes in multi-model ensemble mean drought severity (dimensionless) (i) and model consistency (ii) on a spatial resolution of 0.5$^{o}$
× 0.5$^{o}$, (a) from the baseline period to 1.5$^{o}$C warmer world, (b) from the baseline period to 2$^{o}$C warmer world and (c): (a)-(b). The dark-gray
boxes show the regions adopted by IPCC (2012), which are labeled in (a)(i) using the ID numbers defined in Table 2. Legend in (a)(i) applies to
(b)(i) and (c)(i); legend in (a)(ii) applies to (b)(ii) and (c)(ii).

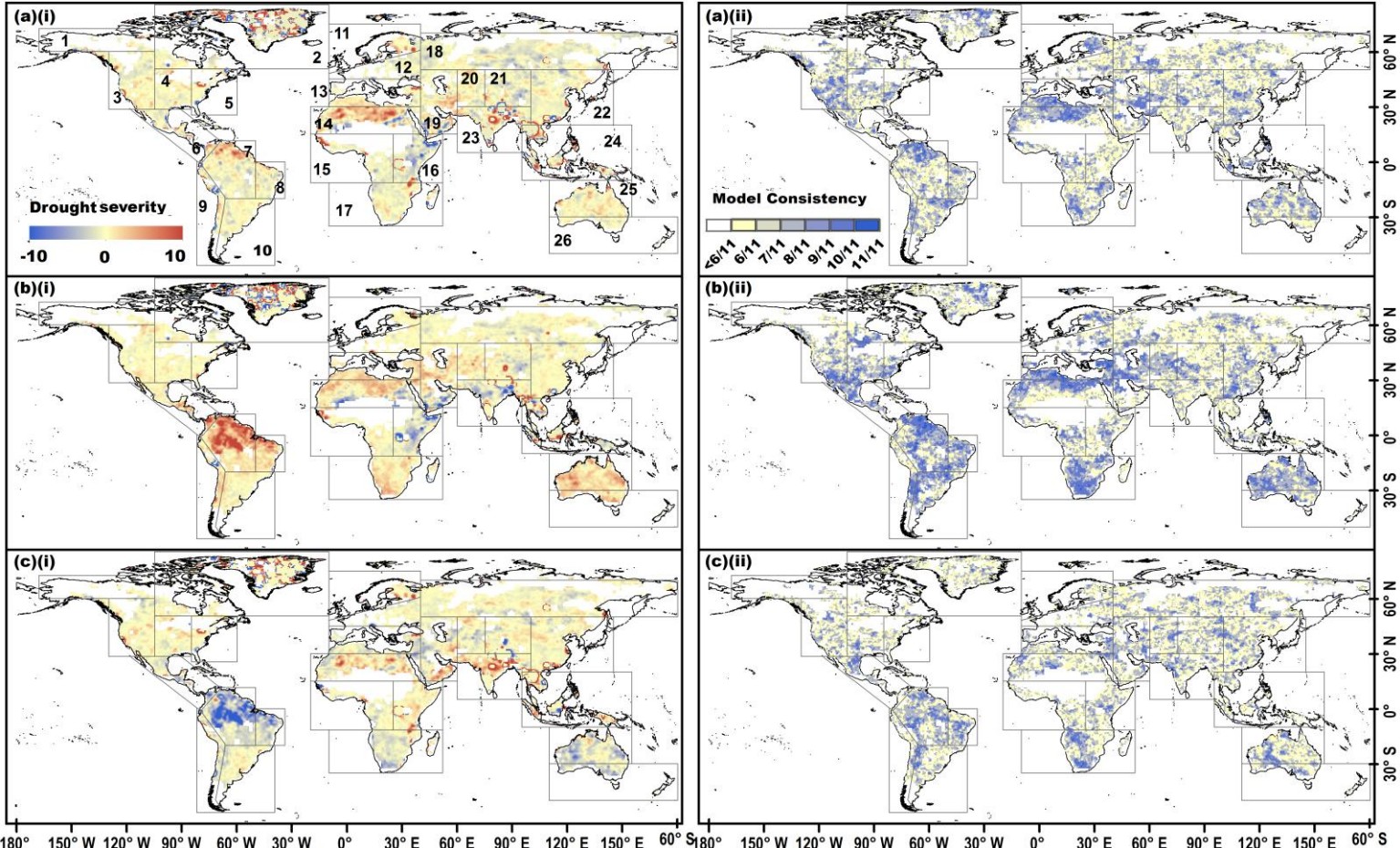


**Figure 10:** Multi-model projected drought severity (dimensionless) at the globe (66°N-66°S) and in 27 regions for the baseline period, 1.5°C and 2°C warmer worlds. The projected uncertainty of multiple climate models is shown through box plots for each region and for each period

**Figure 11:** Multi-model projected frequency (Freq.) and affected total population (Pop., million) of severe drought (PDSI < -3) at the globe and in 27 regions for the baseline period (black, fixed SSP1 2000 population), 1.5$^{o}$C (orange, fixed SSP1 2100 population) and 2$^{o}$C (red, fixed SSP1 2100 population) warmer worlds. The projected uncertainties (standard deviation of multiple-model results) of multiple climate models are shown by error bars (horizontal and vertical)

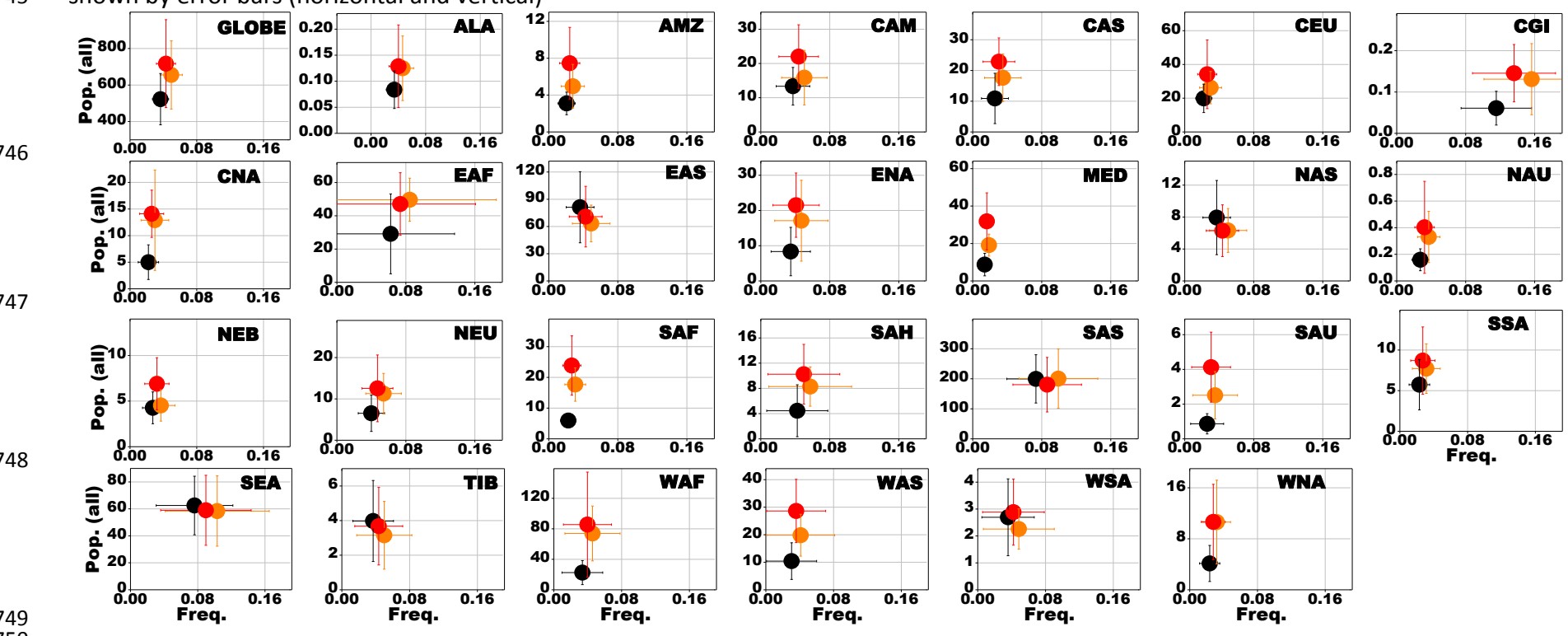

**Figure 12:** Multi-model projected frequency (Freq.) and affected urban population (Pop., million) of severe drought (PDSI < -3) at the globe and in 27 regions for the baseline period (black, fixed SSP1 2000 population), 1.5$^o$C (orange, fixed SSP1 2100 population) and 2$^o$C (red, fixed SSP1 2100 population) warmer worlds. The projected uncertainties (standard deviation of multiple-model results) of multiple climate models are shown by error bars (horizontal and vertical)

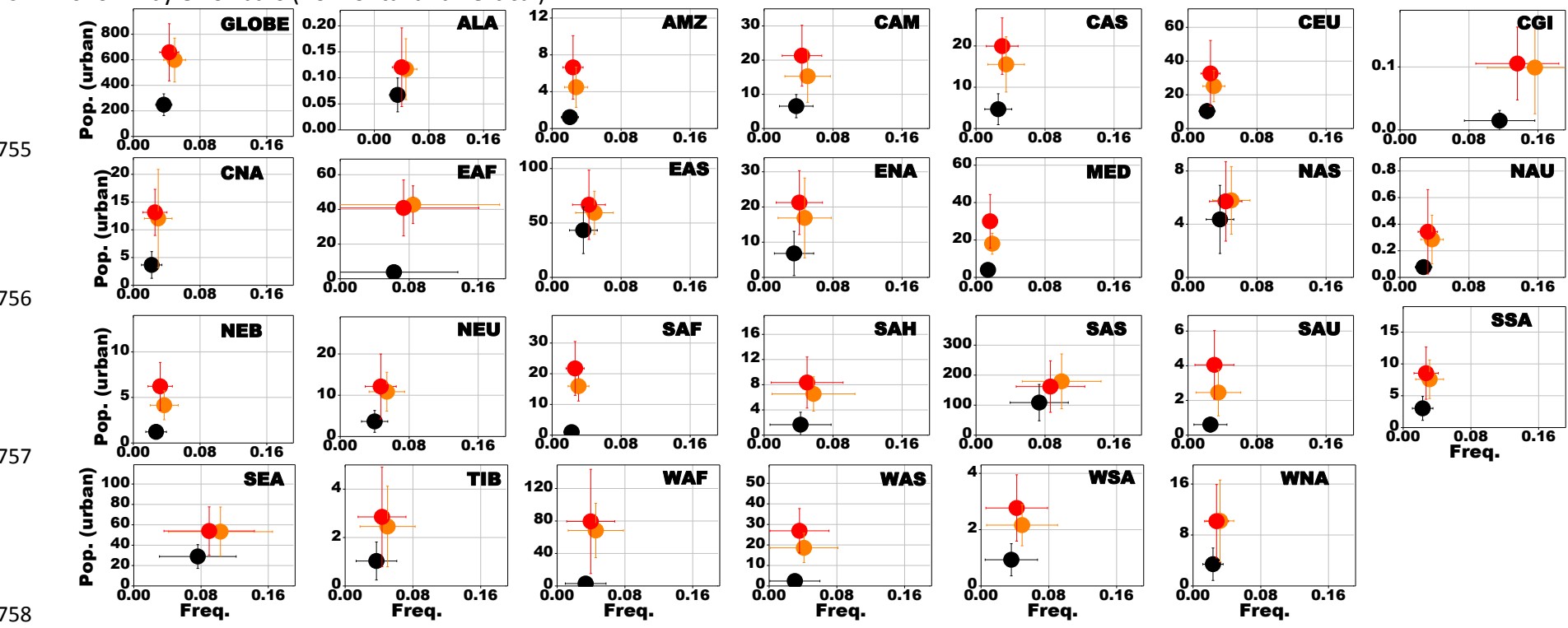

**Figure 13:** Multi-model projected frequency (Freq.) and affected rural population (Pop., million) of severe drought (PDSI < -3) at the globe and in 27 regions for the baseline period (black, fixed SSP1 2000 population), 1.5°C (orange, fixed SSP1 2100 population) and 2°C (red, fixed SSP1 2100 population) warmer worlds. The projected uncertainties (standard deviation of multiple-model results) of multiple climate models are shown by error bars (horizontal and vertical)

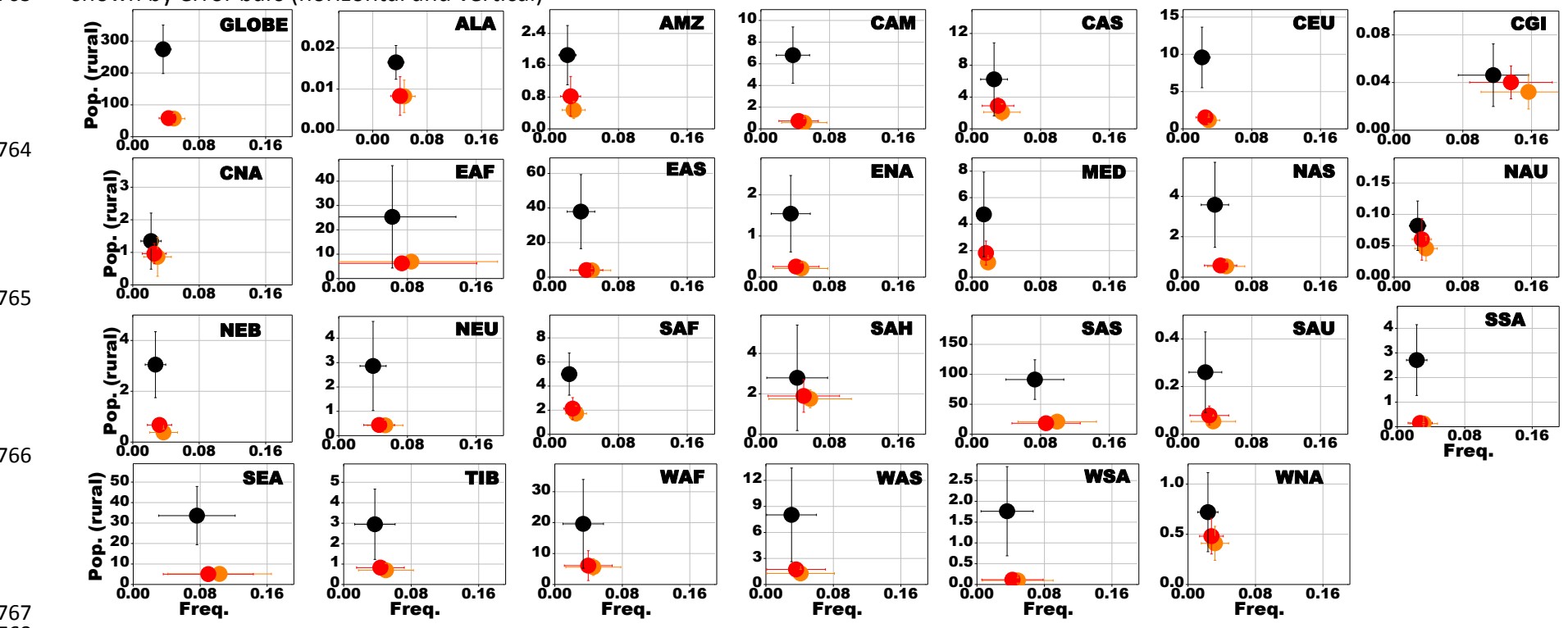