# Peer review of "Global drought and severe drought affected population in 1.5$^{o}$C and 2$^{o}$C warmer worlds"

_Earth System Dynamics, 2017_

## Referee Comment (RC1) · D. Defrance (Referee) · 11 Dec 2017

This article talks about the drought evolution (duration, intensity and frequency) due to the climate change in a 1.5 (2°C) scenarii defined by the COP21. It gives an estimation of the impacted people around the world. To obtain these results, the article uses outputs from eleven CMIP5 GCMs with the RCP4.5 and 8.5, the gridded population from SSP1 scenario and the Palmer drought severity index (PDSI). The transdisciplinary of this article is very interesting and show the human impacts due to the climate change. This paper is divided in 5 parts: an introduction that clearly defines the drought importance on the human society and the 1.5°C (2°C) scenario. The second part describes the data and the method but in this section, some corrections are required (see below). The results are well described and the discussion is interesting but some justifications

could improve the limits of the method. I suggest publishing this article in ESD with major revision. The different remarks and suggestions are described below.

Major comment on the methodology

That is the major comment on your article and the bigger correction I demand. First, I find that you don't give enough details to justify the choose to use eleven-CMIP5 models and only 2 RCP scenarios (4.5 and 8.5). Data are available on about 30 models for RCP8.5 (e.g. Famien et al. ESD discussion November 2017) and available for RCP2.6 and 6 but with a reduced number of models. Are more simulations not better ? I would like a justification of the models use. Secondly, in your study, all models have the same ponderation but if a model is a reverse signal, this result is not visible, to avoid this problem you can use a classification of model type as in Monerie et al, 2016 (10.1007/s00382-016-3236-y) or in Sgubin et al., 2017 (10.1038/ncomms14375.375). I think that the use of a classification is important to improve your results robustness.

My other remark (the more important correction) is due to your impacted study. You write in the discussion that the uncertainty of the model is important and the use of several models weaks the error. That is true but not sufficient. In some region, I think about West Africa, no models have the correct precipitations pattern. This problem is maybe present in other regions. I think this problem lead to a wrong result for the impact and this part must be corrected. The best solution is to unbiase the outputs of the model with e.g. a quantile/quantile method (univariate or multivariate) and observations. These outputs maybe exist now and can improve your interesting results. Another way is better describe the errors between the models and the current observations to be able to determine a confidence index for all regions. With a correction of the output or with a discussion of the local error from the model, the results will be robuster and you can eject the area where the confidence index is not sufficient.

Some questions/remarks

In the abstract: I suggest adding some numerical results from the results and discussion part because of now the abstract is too qualitative.

Line 84: maybe add a reference for the longer duration

Line 101: Why do you use only the SSP1 scenario? That is right this is compatible with the 1.5° scenarios (Line 135) but other SSP scenarios are also compatible with your climatic scenario. Have you an idea of the impacts of the different SSP scenarios on your results?

Line 153: Why do you use +-0.2 around the 1.5°C and 2°C ?

Line 207: Can you define the used threshold to define the drought duration/intensity?

Line 216: That is better to describe a little more the SSP1 scenario for the evolution of the rural/urban people. Your results explain the different trends but we don't know the evolution of the population.

Line 336: I suggest putting this paragraph before the SSP1 results. I think that is more logical to determine the role of the climate on the population exposure with a current population and after you add the role of the demographic trend.

Technical notes

Line 87: (PDSI) in place of ,PDSI,

Line 236: (more drought-prone) ?

Line 280: a "." To delete

---

## Referee Comment (RC2) · Anonymous Referee #2 · 27 Dec 2017

This paper assesses changes in drought risk (and human population exposure to these risks) at 1.5 and 2 degree thresholds drawn from 11 CMIP5 models and the RCP 4.5 and 8.5 scenarios. Unsurprisingly, they find less risk/exposure at 1.5, though the abstract is missing important details of their results (where is mitigation most important for reducing risks?). I think this study has some potential merit, but I have some significant concerns and critiques that I would like to see addressed before I recommend publication.

1) The authors evaluate drought using one specific drought index: PDSI. This is generally fine, but there are some issues regarding how this index is used by the authors. First, despite the title of the original Palmer paper, PDSI is an indicator of agricultural drought risk because it emulates soil moisture availability. Meteorological drought

refers specifically to deficits in precipitation. The language in the paper, and the title, should be adjusted accordingly. Second, it is unclear what time period the authors used for the PDSI calibration (i.e., the CAFEC). Typically, one would use some common historical baseline across models so that future changes can be interpreted relative to historical variability. For this particular study, I would recommend using 1850-2000. Doing it this way would thus not require any differencing between future and historical periods, since the PDSI for the future would implicitly reflect drought changes relative to the historical period. Finally, because of the inherent memory and persistence embedded within the PDSI calculation, this index is much better for picking up long-term and seasonal-scale droughts, and is less appropriate for shorter term (e.g., 1-month) events. For example, the severe and by some indicators record breaking 2012 drought in the Central Plains of the United States only shows up modestly in PDSI, primarily because this drought intensified quite quickly. For this study, where the authors are interested in month to month changes in drought intensity/persistence/etc, it would be better for the authors to use the Z-index that comes out of the PDSI calculation.

2) Given the relative coarseness of the CMIP5 models, I think interpolation of the results to 0.5 degree spatial resolution is not appropriate. A 2 degree common grid would be better, and would avoid effectively making up data at the much finer resolution.

3) The population analysis in this study is a bit convoluted. For example, the RCP scenarios use different populations trajectories (I believe), and since you are picking somewhat arbitrary periods that just match desired warming, there will be little consistency in population structure across either the scenarios or warming targets in this analysis (see Figure 1). Since the 1.5 and 2 degree targets are stabilization scenarios, which would theoretically hold out through the end of the 21st century, I think the authors should remove all the population analyses except for the SSP1 2100 analysis (Figure 4). I would also ask the authors to turn Table 4 into one or more figures, since it is difficult for the reader to synthesize such a large table of numbers.

4) For the analyses, how many datapoints (presumably months) are included in each

warming scenario? What are the units for drought duration (Figure 5), drought intensity (Figure 7), and severity (Figure 9)? Please add this information to the figure captions. Was significance/robustness/consistency only assessed in terms of agreement across the multi-model ensemble (right columns in the aforementioned figures)? If so, what was the threshold used by the authors to determine whether a given change was sufficiently robust?

5) What is causing the changes in drought risk in these simulations? Declines in precipitation or increases in evaporative demand from warming? Since PDSI is an offline calculation, you can recalculate this index using detrended temperature/precipitation to tease this out. This would be a valuable addition.

---

## Author Comment (AC1) · 28 Dec 2017

This article talks about the drought evolution (duration, intensity and frequency) due to the climate change in a 1.5 (2oC) scenarii defined by the COP21. It gives an estimation of the impacted people around the world. To obtain these results, the article uses outputs from eleven CMIP5 GCMs with the RCP4.5 and 8.5, the gridded population from SSP1 scenario and the Palmer drought severity index (PDSI). The transdisciplinary of this article is very interesting and show the human impacts due to the climate change. This paper is divided in 5 parts: an introduction that clearly defines the drought importance on the human society and the 1.5oC (2oC) scenario. The second part describes the data and the method but in this section, some corrections are required (see below). The results are well described and the discussion is interesting but some justifications

could improve the limits of the method. I suggest publishing this article in ESD with major revision. The different remarks and suggestions are described below.

Thank you. In the revised manuscript, we clarified the justifications and methodology of this manuscript. In the acknowledgement section, we added: "We thank the Editor (Michel Crucifix), Dimitri Defrance and another anonymous reviewer for their helpful comments". (Line 486-488 in the latest clean version).

Major comment on the methodology That is the major comment on your article and the bigger correction I demand. First, I find that you don't give enough details to justify the choose to use eleven-CMIP5 models and only 2 RCP scenarios (4.5 and 8.5). Data are available on about 30 models for RCP8.5 (e.g. Famien et al. ESD discussion November 2017) and available for RCP2.6 and 6 but with a reduced number of models. Are more simulations not better? I would like a justification of the models use.

We could have clarified the justification on the selection of climate models and climate scenarios as part of the methodology in the original version of this manuscript. A key step in current study is computation of PDSI using climate model outputs. It requires monthly simulated outputs, i.e., surface mean air temperature, surface minimum air temperature, surface maximum air temperature, air pressure, precipitation, relatively humidity, surface downwelling longwave flux, surface downwelling shortwave flux, surface upwelling longwave flux and surface upwelling shortwave flux; daily simulated outputs, i.e., surface zonal velocity component and meridional velocity component. Whilst a large number of climate models are available in the CMIP5 archive, we made use of those fully satisfied our data requirement. Among the RCP scenarios (i.e., RCP2.6, RCP4.5, RCP6, RCP8.5), the climate models under RCP4.5 and RCP8.5 scenarios are more complete relative to those under RCP2.6 and RCP4.5 scenarios (the reviewer also noticed this). In fact, recent studies have confirmed that the impacts of similar global mean surface temperature (i.e., +1.5oC, +2oC worlds) among the RCP scenarios are quite similar, implying that the global and regional responses to temperature and are independent of the RCP scenarios (Hu et al., 2017; King et al., 2017).

These provide the scientific justification for using the climate models under the RCP4.5 and RCP8.5 scenarios in current study. We settled at using 11 CMIP5 models under these scenarios.

In response, we clarified the justification of climate models and climate scenarios used in Line 129-137, as follows,

"...Recent studies have confirmed that the impacts of similar global mean surface temperature (i.e., +1.5oC and +2oC worlds) among the Representative Concentration Pathways (RCPs) are quite similar, implying that the global and regional responses to temperature and are independent of the RCPs (Hu et al., 2017; King et al., 2017). Following this idea, we settled at using 11 CMIP5 models which satisfied the data requirement of PDSI calculation (see paragraph above) under RCP4.5 and RCP8.5. Following Wang et al. (2017) and King et al. (2017), we use the ensemble mean of these CMIP5 models and climate scenarios (RCP4.5, RCP8.5) to composite the composite the warming scenarios (+1.5oC and +2oC worlds)..."

Reference: King, A.D., Karoly, D.J., and Henley, B.J.: Australian climate extremes at 1.5oC and 2oC of global warming, Nat. Clim. Change, 7, 412-416, doi:10.1038/nclimate3296, 2017.

Schleussner, C., Lissner, T.K., Fischer, E.M., Wohland, J., Perrette, M., Golly, A., Rogelj, J., Childers, K., Schewe, J., Frieler, K., Mengel, M., Hare, W., and Schaeffer, M.: Differential climate impacts for policy-relevant limits to global warming: the case of 1.5oC and 2oC, Earth Syst. Dynam., 7, 327-351, doi: 10.5194/esd-7-327-2016, 2016.

Wang, Z.L., Lin, L., Zhang, X.Y., Zhang, H., Liu, L.K., and Xu, Y.Y.: Scenario dependence of future changes in climate extremes under 1.5oC and 2oC global warming, Sci. Rep., 7:46432, doi:10.1038/srep46432, 2017.

Hu, T., Sun, Y., and Zhang, X.B.: Temperature and precipitation projection at 1.5 and 2oC increase in global mean temperature (in Chinese), Chin. Sci. Bull., 62, 3098-3111,

2017.

Secondly, in your study, all models have the same ponderation but if a model is a reverse signal, this result is not visible, to avoid this problem you can use a classification of model type as in Monerie et al, 2016 (10.1007/s00382-016-3236-y) or in Sgubin et al., 2017 (10.1038/ncomms14375.375). I think that the use of a classification is important to improve your results robustness.

Whilst it is possible to perform classification study, we have concern that it would shift away from the main focus of current study and our target audience (i.e., international policy-makers). This manuscript is prepared to inform climate policy-making hence we generalized the impacts as much as possible. We also performed sufficient uncertainty analyses. For example, we first generalized the multi-model results using the multi-model ensemble mean (the left panels in Figures 3, 5, 7 and 9), as many other studies did (i.e., Mo et al., 2015; He and Soden, 2016). We then quantified the different results/uncertainty among climate models using model consistency (the right panels in Figures 3, 5, 7 and 9) and boxplots (Figures 4, 6, 8 and 10). We characterized different impacts of severe droughts on population at continental scales using the multi-model ensemble mean and the corresponding uncertainty among climate models in Figures 11-13. We also discussed the uncertainties of this study in Section 4 (Line 408-427 in the latest clean version)

Reference: Mo, K.C. and Lyon, B.: Global Meteorological drought prediction using the North American Multi-Model ensemble, J. Hydrometero., 16, 1409-1424, 2015.

He, J., and Soden, B.: A re-examination of the projected subtropical precipitation decline, Nature Climate Change, 17, 53-57

My other remark (the more important correction) is due to your impacted study. You write in the discussion that the uncertainty of the model is important and the use of several models weaks the error. That is true but not sufficient. In some region, I think about West Africa, no models have the correct precipitations pattern. This problem

is maybe present in other regions. I think this problem lead to a wrong result for the impact and this part must be corrected. The best solution is to unbiase the outputs of the model with e.g. a quantile/quantile method (univariate or multivariate) and observations. These outputs maybe exist now and can improve your interesting results. Another way is better describe the errors between the models and the current observations to be able to determine a confidence index for all regions. With a correction of the output or with a discussion of the local error from the model, the results will be robuster and you can eject the area where the confidence index is not sufficient.

We could have explained better about the methodology and the rational in the original version of the manuscript. Please note that the focus of this study is about the change in climate impact/risk (i.e., global meteorological droughts in this study) under +1.5oC and +2oC worlds. In this case, application of bias correction method(s) towards the historical and future periods would be somewhat redundant. The reason is because the change between the bias-corrected results of historical and future is more or less similar to that without bias-correction (e.g., Sun et al., 2011, WRR; Maraun, 2016). In addition, our methodology requires meteorological information (i.e., short- and long-wave radiation, wind, air temperature, humidity, air pressure, precipitation) that is consistent with the energy balance of the climate model (refer to Equation 3 in Section 2.3), hence we have concern about the ability of bias-correction method(s) in maintain the energy balance of the climate models. In terms of confidence index, we rigorously presented model consistency/agreement of these projected changes in climate impact/risk under +1.5oC and 2oC worlds (Figures 3, 5, 7, 9). This kind of confidence index has been widely applied for characterizing multi-model climate projections (e.g., Hirabayashi et al., 2013; Koirala et al., 2014).

In response, we revised Section 2.3 to clarify the rational of our methodology. We also discussed the feasibility using bias-correction approaches and alternative confidence indices (combine with thoughts kindly put forward by the reviewer) in Section 4. Thank you.
Reference: Sun,F., Roderick, M.L., Lim, W.H., and Farquhar, G.D.: Hydroclimatic projections for the Murray-Darling Basin based on an ensemble derived from Intergovernmental Panel on Climate Change AR4 climate models, Water Resource Research, 47, W00G02, 2011

Maraun, D.: Bias correcting climate change simulations – a critical review, Current Climate Change Reports, 2, 211-220, 2016

Hirabayashi, Y., Mahendran, R., Korala, S., Konoshima, L., Yamazaki, D., Watanabe, S., Kim, H. and Kanae, S.: Global flood risk under climate change, Nature Climate Change, 3, 816-821, 2013

Koirala, S., Hirabayashi, Y., Mahendran, R., Kanae, S.: Global assessment of agreement among streamflow projections using CMIP5 model outputs, Environmental Research Letters, 9, 064017 (11pp), 2014

Some questions/remarks In the abstract: I suggest adding some numerical results from the results and discussion part because of now the abstract is too qualitative.

Good point. In response, we added more quantitative results in the abstract (Line 32-45 in the latest clean version). Thank you.

Line 84: maybe add a reference for the longer duration

Done. Thank you.

Reference: McKee, T.B., Doesken, N.J. and Kleist, J.: The relationship of drought frequency and duration of time scales, Eighth Conference on Applied Climatology, American Meteorological Society, Jan 17-23, Anaheim CA, PP. 179-186, 1993.

Line 101: Why do you use only the SSP1 scenario? That is right this is compatible with the 1.5o scenarios (Line 135) but other SSP scenarios are also compatible with your climatic scenario. Have you an idea of the impacts of the different SSP scenarios on your results?

Also refer to our response to the second comment of the reviewer.

We used SSP1 scenario because it describes the storyline of a green growth paradigm with sustainable development and low challenges for adaptation and mitigation (O'Neill et al., 2014, Climatic Change). The +1.5oC and +2oC worlds clearly fit in this description and thus considered under the Paris Agreement 2015 (UNFCCC Conference of the Parties 2015).

Reference: O'Neill et al.: A new scenario framework for climate change research: the concept of shared socioeconomic pathways, Climatic Change, 122, 387-400, 2014. UNFCCC Conference of the Parties 2015: Adoption of the Paris Agreement, FCCC/CP/2015/10Add.11-32Paris

Line 153: Why do you use +-0.2 around the 1.5oC and 2oC ?

This study applied continuous time series for identification of drought duration, intensity and severity. From the climate model projections, we noticed that inter-annual variation of global mean air temperature is common and its magnitude differs with different climate models. To account for it, we followed King et al. (2017), in which the 1.5oC (2oC) world was defined as all years in the 2006-2100 scenario simulations when average temperatures are between 1.3 - 1.7oC (1.8 - 2.2oC) warmer than the pre-industrial levels.

In response, we explained the rational of using +/- 0.2 in Line 157-163.

Reference: King, A.D., Karoly, D.J., and Henley, B.J.: Australian climate extremes at 1.5oC and 2oC of global warming, Nat. Clim. Change, 7, 412-416, doi:10.1038/nclimate3296, 2017.

Line 207: Can you define the used threshold to define the drought duration/ intensity?

We added the used threshold (PDSI< -1) for defining the drought duration/intensity/ severity in the revised manuscript. Thank you.

Line 216: That is better to describe a little more the SSP1 scenario for the evolution of the rural/urban people. Your results explain the different trends but we don't know the evolution of the population.

Good point. In response, we added more descriptions about the evolution of population under the SSP1 scenario as follows (Line 255-262 in the latest clean version),

"…In this pathway, the world population will peak at around 2050s and then decline (van Vuuren et al., 2017). The environmentally friendly living arrangements and human settlement design in this scenario leads to fast urbanization in all countries. More in-migrants from rural areas are attracted to cities due to more adequate infrastructure, employment opportunities and convenient services for their residents (Cuaresma, 2012). The world urban population will gradually increase while rural population will correspondingly decline in the future under SSP1 scenario..."

Line 336: I suggest putting this paragraph before the SSP1 results. I think that is more logical to determine the role of the climate on the population exposure with a current population and after you add the role of the demographic trend.

Done. Thank you.

Technical notes Line 87: (PDSI) in place of, PDSI, Line 236: (more drought-prone)? Line 280: a "." To delete

Done. Thank you.

---

## Editor Comment (EC1) · M. Crucifix (Editor) · 10 Jan 2018

First of all I would like to thank authors and reviewers for their contribution to Earth System Dynamics.

As the authors could read, the reviewers valued the work and consider that it is worthwhile of publication, but they formulate a good number of comments. Some are requests for specification of methodological details, which will be surely satisfied. However, the reviewers also expressed a number of bigger concerns.

Among others,

- Review 1 expresses concerns about systematic biases, and the authors respond

that "In this case, application of bias correction method(s) towards the historical and future periods would be somewhat redundant." I am not convinced by this justification. Both the physics and the impacts of precipitation and drought are highly non-linear.

- Reviewer 2 expresses concerns about resolution, and the authors reply by providing the simulation context which has forced this choice of resolution.

We all fully understand that some methodological choices are forced by the circumstances, techniques and resources available. However the authors consider their paper to be targeted to policy makers. They therefore endorse a role of expert, and this is the expert's job to synthesize the caveats attached to their study, with regard the to possible use of their study for policy decisions. This needs to be done in plain and clear language.

As a side note, the authors state that they "first generalized the multi-model results using the multi-model ensemble mean". The word "generalized" should be replaced by "synthesised", as a multi-ensemble mean is by no mean a generalisation.

The authors are now invited to submit their revised document, which will be sent again to the reviewers.

---

## Author Comment (AC2) · 10 Jan 2018

This paper assesses changes in drought risk (and human population exposure to these risks) at 1.5 and 2 degree thresholds drawn from 11 CMIP5 models and the RCP 4.5 and 8.5 scenarios. Unsurprisingly, they find less risk/exposure at 1.5, though the abstract is missing important details of their results (where is mitigatiodn most important for reducing risks?). I think this study has some potential merit, but I have some significant concerns and critiques that I would like to see addressed before I recommend publication.

Thank you. In the revised manuscript, each of comments/suggestions has been carefully addressed (please find the point to point response below). In the acknowledge-

ment section, we added: "We thank the Editor (Michel Crucifix), Dimitri Defrance and anonymous reviewer for their helpful comments". (Line 462-463 in the latest clean version).

1) The authors evaluate drought using one specific drought index: PDSI. This is generally fine, but there are some issues regarding how this index is used by the authors. First, despite the title of the original Palmer paper, PDSI is an indicator of agricultural drought risk because it emulates soil moisture availability. Meteorological drought refers specifically to deficits in precipitation. The language in the paper, and the title, should be adjusted accordingly.

We accept the criticism.

In response, we replaced "meteorological drought" with "drought" in the title and the main text and adjusted the language (Line 179-180) to reflect the meaning and implication of PDSI as an indicator of drought risk throughout the manuscript. Thank you.

Second, it is unclear what time period the authors used for the PDSI calibration (i.e., the CAFEC). Typically, one would use some common historical baseline across models so that future changes can be interpreted relative to historical variability. For this particular study, I would recommend using 1850-2000. Doing it this way would thus not require any differencing between future and historical periods, since the PDSI for the future would implicitly reflect drought changes relative to the historical period.

We could have explained the calculation of PDSI more clearly in the original version of this manuscript. The parameters (i.e., duration factor) in PDSI model were actually calibrated during the period of 1850-2000 in this study.

In response, we clarified this PDSI calibration and calculation in Section 2.3 (Line 197-198). Thank you.

Finally, because of the inherent memory and persistence embedded within the PDSI calculation, this index is much better for picking up long-term and seasonal-scale
droughts, and is less appropriate for shorter term (e.g., 1-month) events. For example, the severe and by some indicators record breaking 2012 drought in the Central Plains of the United States only shows up modestly in PDSI, primarily because this drought intensified quite quickly. For this study, where the authors are interested in month to month changes in drought intensity/persistence/etc, it would be better for the authors to use the Z-index that comes out of the PDSI calculation.

We agree that the PDSI was criticized for its inability to depict droughts on time scales shorter than 12 month when monthly PDSI values were used (by A.G. Dai, link: https://climatedataguide.ucar.edu/climate-data/palmer-drought-severity-index-pdsi). However, the purpose of this study is to examine the changes in PDSI-detected drought characteristics (i.e., monthly-averaged PDSI, mean drought duration, intensity and severity) between the 1.5/2 degree warming scenario and the baseline period. This time-scale related issue is not the focus of this study. Whilst Z-index could assist short-term drought identification, we are not aware of standardized method(s) from existing literature which could help defining drought duration, intensity and sever-ity. By contrast, the PDSI method appears to be a more compelling method in current study.

2) Given the relative coarseness of the CMIP5 models, I think interpolation of the re-sults to 0.5 degree spatial resolution is not appropriate. A 2 degree common grid would be better, and would avoid effectively making up data at the much finer resolution.

In this study, we first calculated the global PDSI and related drought characteristics (drought duration, intensity and severity) using GCM-outputs with their original spatial resolution (the results were thus not affected by interpolation in this step). The obtained results were then rescaled to a common spatial resolution of $0.5^o \times 0.5^o$ using the bilinear interpolation, in order to (1) show the results with a finer resolution uniformly and (2) accommodate their spatial resolution to that of SSP1 population ($0.5^o \times 0.5^o$). The original resolution of SSP1 population is 0.125 degree. We thus use a 0.5 degree resolution to avoid effectively making up data of the finer resolution in SSP1 data.

In the revised manuscript (Line 227-233), we clarified the justification of spatial resolution used in interpolation as follows, to make them more clear and understandable.

"...It should be noted that the global PDSI and related drought characteristics were first calculated using GCM-outputs with their original spatial resolution. The obtained results were then rescaled to a common spatial resolution of 0.5o ×0.5o using the bilinear interpolation, in order to show them with a finer resolution uniformly and accommodate their spatial resolution to that of SSP1 population (0.5o ×0.5o). The original resolution of SSP1 population is 0.125 degree. We thus use a 0.5 degree resolution to avoid effectively making up data of the finer resolution in SSP1 data..."

3) The population analysis in this study is a bit convoluted. For example, the RCP scenarios use different populations trajectories (I believe), and since you are picking somewhat arbitrary periods that just match desired warming, there will be little consistency in population structure across either the scenarios or warming targets in this analysis (see Figure 1). Since the 1.5 and 2 degree targets are stabilization scenarios, which would theoretically hold out through the end of the 21st century, I think the authors should remove all the population analyses except for the SSP1 2100 analysis (Figure 4). I would also ask the authors to turn Table 4 into one or more figures, since it is difficult for the reader to synthesize such a large table of numbers.

Excellent idea.

In response, we kept the results of SSP1 2100 (SSP1 2100 population for +1.5 and +2 degree warmer scenarios, SSP1 2000 population for the baseline period) and turned them into three figures (Figures 11-13). We removed the old Table 3-4 and Figure 11-13 from the original version of this manuscript. We updated the Method, Results, Conclusion, Abstract sections accordingly. Thank you.

4) For the analyses, how many datapoints (presumably months) are included in each warming scenario?

For the 1.5oC warmer world, there are (12+19)*12=372 data-points (12 years of 2027-2038 under the RCP4.5 scenario and 19 years of 2029-2047 under the RCP8.5 scenario) included in the analyses. For the 2oC warmer world, there are totally (29+12)*12=492 data-points (29 years of 2053-2081 under the RCP4.5 scenario and 12 years of 2042-2053 under the RCP8.5 scenario) included in the analyses.

In response, we added details of this information in Section 2.2 (Line 171-172) of the revised manuscript as follows,

"...It finally results in totally 372 (31 years) and 492 (41 years) data-points (months) for the 1.5oC and 2oC warming scenarios in the following analysis..."

What are the units for drought duration (Figure 5), drought intensity (Figure 7), and severity (Figure 9)? Please add this information to the figure captions.

The unit of drought duration is "months" in this study. Drought intensity and severity are two dimensionless variables.

In response, we added this information to both the figure captions and the main text of the revised manuscript (Line 221-224). Thank you.

Was significance/robustness/consistency only assessed in terms of agreement across the multi-model ensemble (right columns in the aforementioned figures)? If so, what was the threshold used by the authors to determine whether a given change was sufficiently robust?

In this study, we performed significance/robustness/consistency analyses in different ways. We first quantified the robustness of the results among climate models using model consistency (the right panels in Figures 3, 5, 7 and 9) and boxplots (Figures 4, 6, 8 and 10). We then characterized different impacts of severe droughts on population at continental scales using the multi-model ensemble mean and the corresponding uncertainty among climate models in Figures 11-13. For each case, we did not give a fixed threshold to determine whether a given change was sufficiently robust. However,

we gave a range of model consistency (from 6/11 to 11/11) to show the results from non-robust (i.e., <6/11), less-robust, median-robust to sufficiently robust (i.e., 11/11). For example, in the right panel of Figure 3, the robustness of projections increases with higher model consistency and vice-versa.

In response, we revised Lines 403-416 to explain the robustness of these results.

5) What is causing the changes in drought risk in these simulations? Declines in precipitation or increases in evaporative demand from warming? Since PDSI is an offline calculation, you can recalculate this index using detrended temperature/ precipitation to tease this out. This would be a valuable addition.

Whilst it is possible to perform attribution study, we have concern that it would shift away from the main focus of current study (assessing drought risk and its related population impacts under +1.5o and +2.0o warmer worlds) and our target audience (i.e., international policy-makers). There are currently many studies focusing on the changes in drought risk and its attribution (i.e., air temperature, precipitation and potential evaporation) conducted at both global and regional scales and for both historical and future periods (Cook et al., 2014; Ficklin et al., 2015; McCabe and Wolock, 2015; Li et al., 2017). It is an interesting idea worth deep investigation in a separate study, but it is obviously beyond the scope of the current study.

In response, we discussed the plausible future studies related to the review's interesting idea in Line 447-450 as follows,

"...Future studies are needed to obtain a better understanding of the causes of changes in global drought (i.e., decline in precipitation/increase in evaporative demand) under different warming scenarios (1.5 oC and 2 oC), which are very important to the mitigations and adoptions of climate-induced drought risks in the future at both global and regional scales..."

Reference: Li, Z., Chen, Y.N., Fang, G.H., Li, Y.P.: Multivariate assessment and attribution of drought in Central Asia, Sci. Rep., 7, 1316, 2017. McCabe, G.J., Wolock, D.M.: Variability and trend in global drought, Earth and Space Science 2, 223-228, 2015. Ficklin, D.L., Maxwell, J.T., Letsinger, S.L., Gholizadeh, H.: A climate deconstruction of recent drought trends in the United States, Environ. Res. Lett. 10, 044009. Cook, B.I., Smerdon, J.E., Seager, R., Coats, S.: Global warming and 21st century drying. Clim. Dyn. 43(9-10), 2607-2627, 2014.

---

## Author Comment (AC3) · 10 Jan 2018

Attached please find the final revised manuscript (clean version).

Relative to the old manuscript, substantial revisions were made in this version based on the comments/suggestions from the editor and two reviewers. We thank the editor, Dimitri Defrance and anonymous reviewer for their helpful comments.

Please also note the supplement to this comment:
https://www.earth-syst-dynam-discuss.net/esd-2017-85/esd-2017-85-AC3-supplement.pdf

**Supplement:**

**Global drought and severe drought affected population in 1.5°C and 2°C warmer worlds**

Wenbin Liu[a], Fubao Sun[a,b,c,d*], Wee Ho Lim[a,e], Jie Zhang[a], Hong Wang[a],

Hideo Shiogama[f] and Yuqing Zhang[g]

[a] Key Laboratory of Water Cycle and Related Land Surface Processes, Institute of Geographic

Sciences and Natural Resources Research, Chinese Academy of Sciences, Beijing, China

[b] Ecology Institute of Qilian Mountain, Hexi University, Zhangye, China

[c] College of Resources and Environment, University of Chinese Academy of Sciences, Beijing,

China

[d] Center for Water Resources Research, Chinese Academy of Sciences, Beijing, China

[e] Environmental Change Institute, University of Oxford, Oxford, UK

[f] Center for Global Environmental Research, National Institute for Environmental Studies, Tsukuba,

Japan

[g] College of Atmospheric Sciences, Nanjing University of Information Science and Technology,

Nanjing, China

**Pre-submit to**: Earth System Dynamics (*Special issue: Earth System at a 1.5°C warming world*)

**Corresponding to**: Prof. Fubao Sun (*sunfb@igsnrr.ac.cn*), Institute of Geographic Sciences and

Natural Resources Research, Chinese Academy of Sciences

2018/1/10

**Abstract.** In Paris Agreement of 2015, a more ambitious climate change mitigation target, on limiting the global warming at 1.5$^{o}$C instead of 2$^{o}$C above pre-industrial levels, has been proposed. Scientific investigations are necessary to investigate environmental risks associated with these warming targets. This study is the first risk-based assessment of changes in global drought and the impact of severe drought on population at 1.5$^{o}$C and 2$^{o}$C additional warming conditions using the CMIP5 (the fifth Coupled Model Intercomparison Project) climate models. Our results highlight the risk of drought at the globe (drought duration would increase from 2.9 to 3.1~3.2 months) and in several hotspot regions such as Amazon, Northeastern Brazil, South Africa and Central Europe at both 1.5 $^{o}$C and 2 $^{o}$C global warming relative to the historical period. Correspondingly, more total and urban population would be exposed to severe droughts at the globe (+132.5±216.2 million and +194.5±276.5 million total population, +350.2±158.8 million and +410.7±213.5 million urban population for 1.5 $^{o}$C and 2 $^{o}$C warmer scenarios) and some regions (i.e., East Africa, West Africa and South Asia). Meanwhile, less rural population would be exposed to severe drought all over the world under both climate warming and population growth (especially the urbanization-induced population migration). By keeping the warming at 1.5$^{o}$C above the pre-industrial levels instead of 2$^{o}$C, the risks of drought would decrease (i.e., less drought duration, drought intensity and drought severity but relatively more frequent severe drought) and the affected total, urban and rural population would all decrease at global and sub-continental scales. Whilst challenging for both East Africa and South Asia, the benefits of limiting warming to below 1.5$^{o}$C are significant for reducing the risks and societal impacts of global drought.

**1 Introduction**

[revised manuscript text omitted]
. It should be noted that the focus of this study is about the change in climate impact under +1.5$^o$C and +2$^o$C worlds. In this case, application of bias correction towards the historical and future period would be somewhat redundant, because the change between the bias-corrected results of historical and future is more or less similar to that without bias-correction (e.g., Sun et al., 2011; Maraun, 2016). Moreover, our methodology requires meteorological information (i.e., short- and long-wave radation) that is consistent with the energy balance of the climate model (refer to Equation 3), hence we have concern about the ability of bias-correction method(s) in maintain the energy balance of the climate models.

Therefore, we analyze the changes in global drought using the original outputs of climate models in this study.

                <Figure 2, here, thanks>

**2.4 Calculation of population exposure to severe droughts**

Following Wells et al. (2014), we assume that a severe drought event occurs when monthly PDSI

< -3. If a severe drought occurs for at least a month in a year, we would take that year as a severe drought year. For each GCM per period (i.e., the baseline, $1.5^{o}$C and $2^{o}$C warming worlds), we quantify the population (including urban, rural and all population) affected by severe drought per grid-cell as (population × annual frequency of severe drought). We first compute the affected population for the baseline period (1985-2005) using the SSP1 base year (2000). Since the

$1.5^{o}$C and $2^{o}$C targets are stabilization scenarios, which would theoretically hold out through the end of the 21[st] century. We repeat this estimation using the constant SSP1 population data in

2100 for the $1.5^{o}$C and $2^{o}$C warming worlds, which is consistent with the original proposal of Paris

Agreement on stabilizing global warming for the specified targets by end of the 21[st] century. We used SSP1 scenario because it describes the storyline of a green growth paradigm with sustainable development and low challenges for adaptation and mitigation (Jones and O'Niell,

2016). The +$1.5^{o}$C and +$2^{o}$C worlds clearly fit in this description and thus considered under the

Paris Agreement 2015 (UNFCCC Conference of the Parties 2015; O'Niell et al., 2016). In this pathway, the world population will peak at around 2050s and then decline (van Vuuren et al.,

2017). The environmentally friendly living arrangements and human settlement design in this scenario leads to fast urbanization in all countries. More in-migrants from rural areas are attracted to cities due to more adequate infrastructure, employment opportunities and convenient services for their residents (Cuaresma, 2012). The world urban population will gradually increase while rural population will correspondingly decline in the future under SSP1

scenario.

**3 Results**

**3.1 Changes in PDSI and drought characteristics**

We present the changes in multi-model ensemble mean PDSI from the baseline period (1986-2005) to each of the 1.5$^o$C and 2$^o$C scenarios and model consistency in Figure 3. For the

1.5$^o$C warmer world, the PDSI would decrease (more drought-prone) with relatively higher model consistency (6~11 models in totally 11 climate models) in some regions, for example, Amazon (0.7±0.8 -> -0.1±0.2), Northeastern Brazil (0.5±0.6 -> -0.1±0.3), Southern Europe and

Mediterranean (0.4±0.6 -> -0.3±0.2), Central America and Mexico (0.2±0.4 -> -0.2±0.1), Central

Europe (0.3±1.0 -> -0.1±0.4) as well as Southern Africa (0.5±0.5 -> -0.3±0.2); slightly increase (less drought-prone) in Alaska/Northwest Canada (-0.01±0.5 -> -0.3±0.2) and North Asia (-0.1±1.0 ->

-0.2±0.2) but with relatively low model consistency. The geographic pattern of changes in PDSI for the 2$^o$C scenario is quite similar to that of 1.5$^o$C warmer world, but the magnitude of change would intensify (in both direction) East Canada, Greenland, Iceland (-0.3±0.2-> -0.4±0.2), East

Africa (-0.5±0.2-> -0.3±0.2), Northern Europe (-0.3±0.3-> -0.2±0.3), East Asia (-0.3±0.1-> -0.2±0.4),

South Asia (-1.0±1.2-> -0.8±0.3) and West Africa (-0.3±0.2-> -0.3±0.3). When global warming is kept at 1.5$^o$C instead of 2$^o$C above the pre-industrial levels, the PDSI value would be larger at the globe (66$^o$S-66$^o$S, -0.4±0.2-> -0.3±0.2) and most regions (Alaska/Northwest Canada, East Africa,

West Africa, Tibetan Plateau, North Asia, East Asia, South Asia and Southeast Asia) (Figure 4).

                                          <Figure 3, here, thanks>

                                          <Figure 4, here, thanks>

We analyze the changes in drought characteristics such as it duration, severity and intensity under the 1.5$^{o}$C and 2$^{o}$C warming conditions. In terms of the drought duration (Figure 5 and

Figure 6), we find robust large-scale features. For example, the drought duration would generally increase at the globe (2.9±0.5 -> 3.1±0.4 months and 2.9±0.5 -> 3.2±0.5 months from the baseline period to the 1.5$^{o}$C and 2$^{o}$C scenarios) and most regions (especially for Amazon, Sahara,

Northeastern Brazil and North Australia) except for North Asia (2.7±0.6 -> 2.6±0.5 months and

2.7±0.6 -> 2.5±0.4 months) under both the 1.5$^{o}$C and 2$^{o}$C warmer worlds. The high model consistency in most regions (i.e., Amazon, Sahara and Northeastern Brazil) for both warming scenarios gives us more confidence on these projections. Relative to the 2$^{o}$C warmer target, a

1.5$^{o}$C warming target is more likely to reduce drought duration at both global and regional scales (except for Alaska/Northwest Canada, East Africa, Sahara, North Europe, North Asia, South Asia,

Southeast Asia, Tibetan Plateau and West Africa).

                                          <Figure 5, here, thanks>

                                          <Figure 6, here, thanks>

Drought intensity and drought severity are commonly used for quantifying, to what extent, the water availability significantly below normal conditions for a region. In this study, the drought intensity is projected to increase at the globe (0.9±0.3 -> 1.1±0.3 and 0.9±0.3 -> 1.0±0.2 from the baseline period to the 1.5$^{o}$C and 2$^{o}$C scenarios) and in most of the regions except for North Asia,

Southeast Asia and West Africa under the 1.5$^{o}$C and 2$^{o}$C warming scenarios (Figures 7-8).

Compare to the 2$^{o}$C scenario, the drought intensity would obviously be relieved at the global and sub- continental scales except for East Canada, Greenland, Iceland (1.0±0.6 -> 0.8±0.5) and West

North America (0.9±0.3 -> 0.8±0.2) in the 1.5$^o$C warmer world. In addition, the projected drought severity would also increase in both 1.5$^o$C and 2$^o$C warmer worlds at the globe (3.0±1.9 -> 4.5±3.0

and 3.0±1.9 -> 3.8±2.0 from the baseline period to the 1.5$^o$C and 2$^o$C scenarios) and in most regions except for North Asia (1.8±0.6 -> 1.8±0.7 and 1.8±0.6 -> 1.5±0.3) (Figures 9-10). When global warming is maintained at 1.5$^o$C instead of 2$^o$C above the pre-industrial levels, the drought severity would weaken in most regions except for Sahara (3.1±0.9-> 3.5±1.3), North Asia (1.5±0.3

-> 1.8±0.8), Southeast Asia (17.2±20.1 -> 35.8±57.2), and West North America (2.4±1.7 ->

[revised manuscript text omitted]

This study illustrates some of the differences in drought characteristics at both global and sub-continental scales that could be expected in a 1.5$^{\circ}$C and 2$^{\circ}$C warmer worlds. These projections inherited several sources of uncertainty. Firstly, there are considerable uncertainties in the numerical projections from different climate models under varied greenhouse gas emission scenarios, especially on a regional scale (i.e., Sahara, Alaska/Northwest Canada and North Asia).

However, the utility of multiple GCMs and emission scenarios should allow us to generalize future projections than that using single model/scenario (Schleussner et al., 2016; Wang et al., 2017;

[revised manuscript text omitted]

+194.5±276.5 million globally) and urban population (+350.2±158.8 million and +410.7±213.5

million globally) would exposed to severe drought in most regions (especially East Africa, West

Africa and South Asia) for both 1.5 $^o$C and 2 $^o$C warming scenarios, particularly for the latter case.

Meanwhile, less rural population (-217.7±79.2 million and -216.2±82.4 million globally) in, e.g.,

Central Asia, East Canada, Greenland, Iceland, Central North America, Southern Europe and

Mediterranean, North Australia, South Africa, Sahara, South Asia, Tibetan Plateau and West

North America would be affected. When global mean temperature increased by 1.5 $^{o}$C instead of

2 $^{o}$C above the pre-industrial level, the total, urban and rural population affected by severe drought would decline in most regions except for East Africa and South Asia. This means that local governments in East Africa and South Asia should be prepared to deal with drought-driven challenges. Future studies are needed to obtain a better understanding of the causes of changes in global drought (i.e., decline in precipitation/increase in evaporative demand) under different warming scenarios (1.5 $^{o}$C and 2 $^{o}$C), which are very important to the mitigations and adoptions of climate-induced drought risks in the future at both global and regional scales.

[revised manuscript text omitted]

---

## Author Response (AR1)

**Response to the Editor**

First of all I would like to thank authors and reviewers for their contribution to Earth System Dynamics. As the authors could read, the reviewers valued the work and consider that it is worthwhile of publication, but they formulate a good number of comments. Some are requests for specification of methodological details, which will be surely satisfied. However, the reviewers also expressed a number of bigger concerns.

Thank you. In the revised manuscript, each of comments has been carefully addressed (*please find the point to point response below*). In the acknowledgement section, we added: "*We thank the Editor (Michel Crucifix), Dimitri Defrance and an anonymous reviewer for their helpful comments*". (*Line 452-453 in the latest clean version*).

Among others,
• Review 1 expresses concerns about systematic biases, and the authors respond in discussion paper that "In this case, application of bias correction method(s) towards the historical and future periods would be somewhat redundant." I am not convinced by this justification. Both the physics and the impacts of precipitation and drought are highly non-linear.

We would like to clarify our point. It is common to find systematic biases in climate model output, where application of bias-correction measures would be helpful for adjusting the "*absolute value*" of a target climate model output (*e.g., precipitation or temperature*) so that its statistics (*e.g., distribution*) would be consistent with that of the observations. Nonetheless, recent climate change studies have investigated and demonstrated that the calculated "*change*" (*between the historical and future periods*) with and without bias-correction is quite similar (*e.g., Sun et al., 2011; Maraun, 2016*). This is because similar statistical adjustment made to the historical and future periods cancelled off one another. In addition, the climate model outputs are constrained by energy balance (*i.e., radiative, latent heat, sensible heat fluxes and heat storage of the earth system*). Whilst we are confident that the application of raw climate model output (*short- and long-wave radiation, air temperature, wind, humidity, atmospheric pressure, precipitation*) as part of our methodology (*Equation 3 in Section 2.3*) is consistent with the energy balance of a climate model, we not convinced enough to use existing bias correction measures To our knowledge, existing bias correction measures do not explicitly considers the energy balance of climate model output. We are also not quite sure whether the physical relationships among different meteorological variables (*i.e., air temperature, precipitation, humidity*) used can still be hold after bias correction. For these reasons, we make use of the raw climate model output in current analysis (*please also see our response to the third major comment of reviewer 1*).

We think that future innovation with proper account for both statistics and energy balance (*and also the multivariable relationships*) of climate model output in new bias-correction methodology for handling the highly non-linear outcomes (*as the Editor noticed*) should be a subject of scientific interest. Because of this possibility, we removed the phrase "*…would be somewhat redundant …*" from our earlier response.

In response, we revised Section 2.3 to clarify the rationale of our methodology. We also discussed the feasibility of using existing bias correction measures and point out that future innovation in bias correction methodology should be a subject of scientific interest in Section 4.

*Reference*:
Sun,F., Roderick, M.L., Lim, W.H., and Farquhar, G.D.: Hydroclimatic projections for the Murray-Darling Basin based on an ensemble derived from Intergovernmental Panel on Climate Change AR4 climate models, Water Resource Research, 47, W00G02, 2011
Maraun, D.: Bias correcting climate change simulations – a critical review, Curr. Clim. Change Rep., 2, 211-220, 2016

• Reviewer 2 expresses concerns about resolution, and the authors reply by providing the simulation context which has forced this choice of resolution.
Please find our response to Reviewer 2 in the latter part of this word document. Thank you.

We all fully understand that some methodological choices are forced by the circumstances, techniques and resources available. However the authors consider their paper to be targeted to policy makers. They therefore endorse a role of expert, and this is the expert's job to synthesize the caveats attached to their study, with regard to possible use of their study for policy decisions. This needs to be done in plain and clear language.
In the revised manuscript, we synthesized helpful feedbacks from the editor and reviewers into the manuscript. We used plain language throughout the manuscript (*please see the marked version*) in order to convey clear scientific findings for the policymakers to develop informed decision. Thank you.

As a side note, the authors state that they "first generalized the multi-model results using the multi-model ensemble mean". The word "generalized" should be replaced by "synthesised", as a multi-ensemble mean is by no mean a generalisation.
Good idea. In response, we replaced the word "*generalized*" with "*synthesized*" wherever appropriate in the revised manuscript and our response to reviewers. Thank you..

The authors are now invited to submit their revised document, which will be sent again to the reviewers.
Thank you.

**Response to review comments - RC1:**

This article talks about the drought evolution (duration, intensity and frequency) due to the climate change in a 1.5 ($2^{o}$C) scenarii defined by the COP21. It gives an estimation of the impacted people around the world. To obtain these results, the article uses outputs from eleven CMIP5 GCMs with the RCP4.5 and 8.5, the gridded population from SSP1 scenario and the Palmer drought severity index (PDSI). The transdisciplinary of this article is very interesting and show the human impacts due to the climate change. This paper is divided in 5 parts: an introduction that clearly defines the drought importance on the human society and the 1.5$^{o}$C ($2^{o}$C) scenario. The second part describes the data and the method but in this section, some corrections are required (see below). The results are well described and the discussion is interesting but some justifications could improve the limits of the method. I suggest publishing this article in ESD with major revision. The different remarks and suggestions are described below.

Thank you. In the revised manuscript, we clarified the justifications and methodology of this manuscript. In the acknowledgement section, we added: "*We thank the Editor (Michel Crucifix), Dimitri Defrance and an anonymous reviewer for their helpful comments*". (*Line 452-453 in the latest clean version*).

**Major comment on the methodology**

That is the major comment on your article and the bigger correction I demand. First, I find that you don't give enough details to justify the choose to use eleven-CMIP5 models and only 2 RCP scenarios (4.5 and 8.5). Data are available on about 30 models for RCP8.5 (e.g. Famien et al. ESD discussion November 2017) and available for RCP2.6 and 6 but with a reduced number of models. Are more simulations not better? I would like a justification of the models use.

We could have clarified the justification on the selection of climate models and climate scenarios as part of the methodology in the original version of this manuscript. A key step in current study is computation of PDSI using climate model outputs. It requires monthly simulated outputs, i.e., surface mean air temperature, surface minimum air temperature, surface maximum air temperature, air pressure, precipitation, relatively humidity, surface downwelling longwave flux, surface downwelling shortwave flux, surface upwelling longwave flux and surface upwelling shortwave flux; daily simulated outputs, i.e., surface zonal velocity component and meridional velocity component. Whilst a large number of climate models are available in the CMIP5 archive, we made use of those fully satisfied our data requirement. Among the RCP scenarios (*i.e., RCP2.6, RCP4.5, RCP6, RCP8.5*), the climate models under RCP4.5 and RCP8.5 scenarios are more complete relative to those under RCP2.6 and RCP4.5 scenarios (the reviewer also noticed this). In fact, recent studies have confirmed that the impacts of similar global mean surface temperature *(i.e., +1.5$^{o}$C, +2$^{o}$C worlds)* among the RCP scenarios are quite similar, implying that the global and regional responses to temperature and are independent of the RCP scenarios (*Hu et al., 2017; King et al., 2017*). These provide the scientific justification for using the climate models under the RCP4.5 and RCP8.5 scenarios in current study. We settled at using 11 CMIP5 models under these scenarios.

In response, we clarified the justification of climate models and climate scenarios used in Line 128-135, as follows,

*"…Recent studies have confirmed that the impacts of similar global mean surface temperature (i.e., 1.5$^o$C and 2$^o$C warmer worlds) among the Representative Concentration Pathways (RCPs) are quite similar, implying that the global and regional responses to temperature and are independent of the RCPs (Hu et al., 2017; King et al., 2017). Following this idea, we settled at using 11 CMIP5 models which satisfied the data requirement of PDSI calculation (see paragraph above) under RCP4.5 and RCP8.5. Following Wang et al. (2017) and King et al. (2017), we use the ensemble mean of these CMIP5 models and climate scenarios (RCP4.5, RCP8.5) to composite the warming scenarios (1.5$^o$C and 2$^o$C warmer worlds)…"*

My other remark (the more important correction) is due to your impacted study. You write in the discussion that the uncertainty of the model is important and the use of several models weaks the error. That is true but not sufficient. In some region, I think about West Africa, no models have the correct precipitations pattern. This problem is maybe present in other regions. I think this problem lead to a wrong result for the impact and this part must be corrected. The best solution is to unbiase the outputs of the model with e.g. a quantile/quantile method (univariate or multivariate) and observations. These outputs maybe exist now and can improve your interesting results. Another way is better describe the errors between the models and the current observations to be able to determine a confidence index for all regions. With a correction of the output or with a discussion of the local error from the model, the results will be robuster and you can eject the area where the confidence index is not sufficient.

We could have explained better about the methodology and the rational in the original version of the manuscript. It is common to find systematic biases in climate model output, where application of bias-correction measures would be helpful for adjusting the "*absolute value*" of a target climate model output (*e.g., precipitation or temperature*) so that its statistics (*e.g., distribution*) would be consistent with that of the observations. Nonetheless, recent climate change studies have investigated and demonstrated that the calculated "*change*" (*between the historical and future periods*) with and without bias-correction is quite similar (*e.g., Sun et al., 2011; Maraun, 2016*). This is because similar statistical adjustment made to the historical and future periods cancelled off one another. In addition, the climate model outputs are constrained by energy balance (*i.e., radiative, latent heat, sensible heat fluxes and heat storage of the earth system*). Whilst we are confident that the application of raw climate model output (*short- and long-wave radiation, air temperature, wind, humidity, atmospheric pressure, precipitation*) as part of our methodology (*Equation 3 in Section 2.3*) is consistent with the energy balance of a climate model, we not convinced enough to use existing bias correction measures To our knowledge, existing bias correction measures do not explicitly considers the energy balance of climate model output. We are also not quite sure whether the physical relationships among different meteorological variables (*i.e., air temperature, precipitation, humidity*) used can still be hold after bias correction. For these reasons, we make use of the raw climate model output in current analysis (*please also see our response to the first comment of the Editor*).

In response, we revised Section 2.3 to clarify the rational of our methodology. We also discussed the feasibility using bias-correction approaches and alternative confidence indices (*combine with thoughts kindly put forward by the reviewer*) in Section 4. Thank you.

Line 336: I suggest putting this paragraph before the SSP1 results. I think that is more logical to determine the role of the climate on the population exposure with a current population and after you add the role of the demographic trend.

Done. Thank you.

**Technical notes**
Line 87: (PDSI) in place of, PDSI,
Line 236: (more drought-prone)?
Line 280: a "." To delete

Done. Thank you.

**Response to review comments – RC2:**

This paper assesses changes in drought risk (and human population exposure to these risks) at 1.5 and 2 degree thresholds drawn from 11 CMIP5 models and the RCP 4.5 and 8.5 scenarios. Unsurprisingly, they find less risk/exposure at 1.5, though the abstract is missing important details of their results (where is mitigatiodn most important for reducing risks?). I think this study has some potential merit, but I have some significant concerns and critiques that I would like to see addressed before I recommend publication.

Thank you. In the revised manuscript, each of comments/suggestions has been carefully addressed (*please find the point to point response below*). In the acknowledgement section, we added: "*We thank the Editor (Michel Crucifix), Dimitri Defrance and an anonymous reviewer for their helpful comments*". (*Line 452-453 in the latest clean version*).

1) The authors evaluate drought using one specific drought index: PDSI. This is generally fine, but there are some issues regarding how this index is used by the authors. First, despite the title of the original Palmer paper, PDSI is an indicator of agricultural drought risk because it emulates soil moisture availability. Meteorological drought refers specifically to deficits in precipitation. The language in the paper, and the title, should be adjusted accordingly.

We accept the criticism.

In response, we replaced "*meteorological drought*" with "*drought*" in the title and the main text and adjusted the language (*Line 176-179*) to reflect the meaning and implication of PDSI as an indicator of drought risk throughout the manuscript. Thank you.

Second, it is unclear what time period the authors used for the PDSI calibration (i.e., the CAFEC). Typically, one would use some common historical baseline across models so that future changes can be interpreted relative to historical variability. For this particular study, I would recommend using 1850-2000. Doing it this way would thus not require any differencing between future and historical periods, since the PDSI for the future would implicitly reflect drought changes relative to the historical period.

We could have explained the calculation of PDSI more clearly in the original version of this manuscript. The parameters (i.e., *duration factor*) in PDSI model were actually calibrated during the period of 1850-2000 in this study.

In response, we clarified this PDSI calibration and calculation in Section 2.3 (*Line 195-196*). Thank you.

Finally, because of the inherent memory and persistence embedded within the PDSI calculation, this index is much better for picking up long-term and seasonal-scale droughts, and is less appropriate for shorter term (e.g., 1-month) events. For example, the severe and by some indicators record breaking 2012 drought in the Central Plains of the United States only shows up modestly in PDSI, primarily because this drought intensified quite quickly. For this study, where the authors are interested in month to month changes in drought intensity/persistence/etc, it would be better for the authors to use the Z-index that comes out of the PDSI calculation.

We agree that the PDSI was criticized for its inability to depict droughts on time scales shorter than 12 month when monthly PDSI values were used (*by A.G. Dai, link*: https://climatedataguide.ucar.edu/climate-data/palmer-drought-severity-index-pdsi). However, the purpose of this study is to examine the changes in PDSI-detected drought characteristics (*i.e., monthly-averaged PDSI, mean drought duration, intensity and severity*) between the 1.5/2 degree warmer world and the baseline period. This time-scale related issue is not the focus of this study. Whilst Z-index could assist short-term drought identification, we are not aware of standardized method(s) from existing literature which could help defining drought duration, intensity and severity. By contrast, the PDSI method appears to be a more compelling method in current study.

2) Given the relative coarseness of the CMIP5 models, I think interpolation of the results to 0.5 degree spatial resolution is not appropriate. A 2 degree common grid would be better, and would avoid effectively making up data at the much finer resolution.

In this study, we first calculated the global PDSI and related drought characteristics (*drought duration, intensity and severity*) using GCM-outputs with their original spatial resolution (*the results were thus not affected by interpolation in this step*). The obtained results were then rescaled to a common spatial resolution of $0.5^o \times 0.5^o$ using the bilinear interpolation, in order to (1) show the results with a finer resolution uniformly and (2) accommodate their spatial resolution to that of SSP1 population ($0.5^o \times 0.5^o$). The original resolution of SSP1 population is 0.125 degree. We thus use a 0.5 degree resolution to avoid effectively making up data of the finer resolution in SSP1 data.

In the revised manuscript (*Line 225-231*), we clarified the justification of spatial resolution used in interpolation as follows, to make them more clear and understandable.

"*…It should be noted that the global PDSI and related drought characteristics were first calculated using GCM-outputs with their original spatial resolution. The obtained results were then rescaled to a common spatial resolution of $0.5^o \times 0.5^o$ using the bilinear interpolation, in order to show them with a finer resolution uniformly and accommodate their spatial resolution to that of SSP1 population ($0.5^o \times 0.5^o$). The original resolution of SSP1 population is 0.125 degree. We thus use a 0.5 degree resolution to avoid effectively making up data of the finer resolution in SSP1 data…*"

3) The population analysis in this study is a bit convoluted. For example, the RCP

scenarios use different populations trajectories (I believe), and since you are picking somewhat arbitrary periods that just match desired warming, there will be little consistency in population structure across either the scenarios or warming targets in this analysis (see Figure 1). Since the 1.5 and 2 degree targets are stabilization scenarios, which would theoretically hold out through the end of the 21st century, I think the authors should remove all the population analyses except for the SSP1 2100 analysis (Figure 4). I would also ask the authors to turn Table 4 into one or more figures, since it is difficult for the reader to synthesize such a large table of numbers. Excellent idea.

In response, we kept the results of SSP1 2100 (*SSP1 2100 population for +1.5 and +2 degree warmer scenarios, SSP1 2000 population for the baseline peri*od) and turned them into three figures (*Figures 11-13*). We removed the old Table 3-4 and Figure 11-13 from the original version of this manuscript. We updated the Method, Results, Conclusion, Abstract sections accordingly. Thank you.

4) For the analyses, how many datapoints (presumably months) are included in each warming scenario?
For the 1.5$^o$C warmer world, there are (12+19)*12=372 data-points (*12 years of 2027-2038 under the RCP4.5 scenario and 19 years of 2029-2047 under the RCP8.5 scenario*) included in the analyses. For the 2$^o$C warmer world, there are totally (29+12)*12=492 data-points (*29 years of 2053-2081 under the RCP4.5 scenario and 12 years of 2042-2053 under the RCP8.5 scenario*) included in the analyses.

In response, we added details of this information in Section 2.2 (*Line 169-170*) of the revised manuscript as follows,

*"…In the 1.5$^o$C and 2$^o$C warmer worlds, we get 372 and 492 monthly data-points, respectively...."*

What are the units for drought duration (Figure 5), drought intensity (Figure 7), and severity (Figure 9)? Please add this information to the figure captions.
The unit of drought duration is "*months*" in this study. Drought intensity and severity are two dimensionless variables.

In response, we added this information to both the figure captions and the main text of the revised manuscript (*Line 219-223*). Thank you.

Was significance/robustness/consistency only assessed in terms of agreement across the multi-model ensemble (right columns in the aforementioned figures)? If so, what was the threshold used by the authors to determine whether a given change was sufficiently robust?
In this study, we performed significance/robustness/consistency analyses in different ways. We first quantified the robustness of the results among climate models using model consistency (*the right panels in Figures 3, 5, 7 and 9*) and boxplots (*Figures 4, 6, 8 and 10*). We then characterized different impacts of severe droughts on population at continental scales using the multi-model ensemble mean and the corresponding uncertainty among climate models in Figures 11-13. For each case, we did not give a fixed threshold to determine whether a given change was sufficiently robust. However, we gave a range of model consistency (from 6/11 to 11/11) to show the results from non-robust (i.e., <6/11), less-robust, median-robust to sufficiently robust (i.e., 11/11). For example, in the right panel of Figure 3, the robustness of projections increases with higher model consistency and vice-versa.

In response, we revised Lines 388-402 to explain the robustness of these results.

5) What is causing the changes in drought risk in these simulations? Declines in precipitation or increases in evaporative demand from warming? Since PDSI is an offline calculation, you can recalculate this index using detrended temperature/ precipitation to tease this out. This would be a valuable addition.

Whilst it is possible to perform attribution study, we have concern that it would shift away from the main focus of current study (*assessing drought risk and its related population impacts under +1.5$^o$ and +2.0$^o$ warmer worlds*) and our target audience (*i.e., international policy-makers*). There are currently many studies focusing on the changes in drought risk and its attribution (*i.e., air temperature, precipitation and potential evaporation*) conducted at both global and regional scales and for both historical and future periods (*Cook et al., 2014; Ficklin et al., 2015; McCabe and Wolock, 2015; Li et al., 2017*). It is an interesting idea worth deep investigation in a separate study, but it is obviously beyond the scope of the current study.

In response, we discussed the plausible future studies related to the review's interesting idea in Line 437-440 as follows,

*"…Future studies on understanding the causes of changes in global and regional droughts (e.g., changing pattern/duration of precipitation and evaporative demand) with respect to these warming targets should assist drought risk adaptation and mitigation planning..."*


King et al. (2017), we use the ensemble mean of these CMIP5 models and climate scenarios (RCP4.5, RCP8.5) to composite the warming scenarios (+1.5oC and +2oC worlds).. Some studies (King et al., 2017; Hu et al., 2017) have found that the global and regional responses (i.e., warming/precipitation patterns) to varied warming scenarios (i.e., 1.5oC and 2oC) showed little dependences on RCP scenarios (RCP2.6, RCP6, RCP4.5 and RCP8.5). Therefore, in this study, following the approaches adopted by Wang et al. (2017) and King et al. (2017), we only chose two RCP scenarios to composite the warming scenarios of 1.5oC and 2oC using the ensemble mean of multiple climate models and emission scenarios. Based on data availability, we select 11

GCMs to perform the analysis (see details of these GCMs in Table 1). In the CMIP5 archive, the monthly uwnd and vwnd were computed as the means of their daily values with the plus-minus sign, the calculated wind speed from the monthly uwnd and vwnd would be equal to or, in most cases, less than that computed from the daily values (Liu and Sun, 2016). To get the monthly wind speed, we average the daily values ($\sqrt{uwnd^2 + vwnd^2}$) over a month. We rescale all data to a common spatial resolution of $0.5° \times 0.5°$ using the bilinear interpolation.

                <Table 1, here, thanks>

To consider the people affected by severe drought events, we use the spatial explicit global population scenarios developed by researchers from the Integrated Assessment Modeling (IAM)

group of National Center for Atmospheric Research (NCAR) and the City University of New York

Institute for Demographic Research (Jones and O'Neil, 2016). They included the gridded population data for the baseline year (2000) and for the period of 2010-2100 in ten-year steps at a spatial resolution of 0.125 degree, which are consistent with the new Shared Socioeconomic

Pathways (SSPs). We apply the population data of the SSP1 scenario, which describes a future pathway with sustainable development and low challenges for adaptation and mitigation. We upscale this product to a spatial resolution of $0.5^o \times 0.5^o$. For the global and sub-continental scales analysis, we use the global land mass between $66^oN$ and $66^oS$ (Fischer et al., 2013; Schleussner et al., 2016) and 26 sub-continental regions (as used in IPCC, 2012, see Table 2 for details).

     <Table 2, here, thanks>

**2.2 Definition of a baseline, 1.5$^o$C and 2$^o$C warmer worlds**

To define a baseline, 1.5$^o$C and 2$^o$C warmer worlds, we first calculate the global mean surface air temperature (GMT) for each climate model and emission scenario over the period 1850-2100.

We weighnote that the surface air temperature field need be weighted by the square root of cosine (latitude) to consider the dependence of grid density on latitude (Liu et al., 2016). The We compute and smooth the Mmulti-model Ensemble Mean (MEM) GMT were computed and smoothed 
[revised manuscript text omitted]

<Figure 2, here, thanks>

**2.4 Calculation of population exposure underto severe droughts**

Following Wells et al. (2014), we assume that a severe drought event occurs when monthly PDSI

< -3, we assume a severe drought event took place. If a severe drought occurredoccurs for at least a month in a year, we would take that year —as a severe drought year. For each GCM per period (i.e., the baseline, 1.5$^{o}$C and 2$^{o}$C warmering worlds), we quantify the population (including urban, rural and all population) affected by severe drought per grid-cell as (population × annual frequency of severe drought). We first compute the affected population for the baseline period (1985-2005) using the SSP1 base year (2000). Since the 1.5$^{o}$C and 2$^{o}$C targets are stabilization scenarios, which would theoretically hold out through the end of the 21$^{st}$ century.

[revised manuscript text omitted]

88% of the affected urban population as well as 114% and -67% of the affected rural population)

are attributable to population growth while others are solely due to climate change impact under the 1.5$^o$C and 2$^o$C warming scenarios, respectively. The climate change driven severe drought affected total, urban and rural population would decrease in East Africa and South Asia but increase (the climate change driven severe drought affected population constitutes >50% of that considering both climate change and population growth) in most regions (for example, North

Australia, Southern Europe and Mediterranean, Central Europe, North Europe, West Coast South

America, Northeastern Brazil, Central North America and East Asia for total population as well as

West North America, Central North America, East North America, Central America and Mexico,

Amazon, Northeastern Brazil, West Coast South America, Southeastern South America, Northern

Europe, Central Europe, Southern Europe and Mediterranean, West Africa, South Africa, North

Asia, Central Asia, Tibetan Plateau, East Asia and Southeast Asia for rural population) under these warming targets.

<Figure 9, here, thanks>

<Figure 10, here, thanks>

Compare to the baseline period, the frequency of severe drought (PDSI < -3), drought-affected total and urban population would increase in most of the regions under the 1.5°C and 2°C

warming scenarios.

<Figure 9, here, thanks>

<Figure 10, here, thanks>

The projections suggest that more urban population would   expose to severe drought in Central

Europe (16.4±8.5 million), Southern Europe and Mediterranean (13.3±4.7 million), West Africa (26.7±15.7 million), East Asia (44.3±20.6 million), South Asia (17.7±37.6 million), West Asia (14.9±7.8 million) and Southeast Asia (19.9±17.0 million) in the 1.5°C warmer world relative to the baseline period. We also find that the number of affected people would escalate further in these regions under the 2°C warming scenario (Table 3). In terms of the rural population, more people in Central Asia (2.2±6.4 million and 1.2±3.5 million for the 1.5°C and 2°C warmer worlds),

Central North America (0.8±1.7 million and 0.4±1.0 million), Southern Europe and Mediterranean (0.9±3.4 million and 0.3±4.9 million), South Africa (1.9±1.9 million and 0.9±2.6 million), Sahara (0.1±2.5 million and 0.3±3.2 million), South Asia (45.9±52.4 million and  23.1±26.9 million),

Tibetan Plateau (0.3±2.7 million and  1.03±2.1 million) and West North America ( 1.2±0.9 million

and 0.1±0.1 million) would expose to the severe drought in the 1.5°C and 2°C warmer worlds relative to the baseline period (except for the globe, South Asia and Tibetan Plateau under the

2°C warming scenario and West North America under the 1.5°C warming scenario). Overall, the multi-model projected uncertainty of affected population (including total, urban and rural population) is quite small in most regions except for East Asia, South Asia and West Africa.

                <Figure 11, here, thanks>

                <Figure 12, here, thanks>

When global warming approaches 1.5°C (instead of 2°C) above the pre-industrial levels, relatively less total (except for total population affected in North Asia, East Asia, Southeast Asia and South

Asia) and urban population would be affected despite more frequent severe drought. By contrast, the affected rural population would increase in most regions (except for Amazon, East Canada,

Greenland, Iceland, North Australia, Northeastern Brazil, Sahara and South Australia/New

Zealand). This implies that the benefit of holding global warming at 1.5$^{o}$C instead of 2$^{o}$C is apparent to the severe-drought affected total and , urban and rural population in most regions, but challenges remain in the rural areas (especially in South Asia, Southeast Asia, East Africa and

West Africa)East Africa and South Asia.

                <Figure 13, here, thanks>

                <Table 3, here, thanks>

We repeat similar analysis using the constant SSP1 population in 2100 (Figure 4). Relative to the baseline period, we find that 38.6±272.7million (38.7±247.1 million urban population and

0.0±26.1million rural population) and 100.3±323.9 million (99.1±295.9 million urban population and 1.5±28.5 million rural population) additional population would expose to severe droughts in the 1.5°C and 2°C warmer worlds on a global scale. The severe drought affected total, urban and rural population would increase under these warming targets in most regions except for Sahara (total and rural population under 1.5°C warming scenario and rural population under 2°C warming scenario), East Africa, South Asia and West Africa (rural population under 1.5°C warming scenario). Moreover, compare to the 2°C warming target, the severe drought affected total, urban and rural population would decrease in the 1.5°C warmer world in most regions such as East Asia, Southern Europe and Mediterranean, Central Europe and Amazon.

To exclude the role of population growth in the former analysis, we also repeat the analysis but this time we keep the population constant in 2000 (Table 4). Globally, we estimate that 63.8±195.9 million (33.3±86.6 million urban population and 30.5±113.6 million rural population) and 122.4±249.9 million (55.4±101.8 million urban population and 67.0±150.7 million rural population) additional people would be exposed solely to severe droughts in the 1.5°C and 2°C warmer worlds, respectively. In terms of percentage, about 75% and 50% of total affected population (considers both severe droughts and population growth, including 86% and 88% of the affected urban population as well as 114% and -67% of the affected rural population) are attributable to population growth while others are solely due to climate change impact under the 1.5°C and 2°C warming scenarios, respectively. The climate change driven severe drought affected total, urban and rural population would decrease in East Africa and South Asia but increase (the climate change driven severe drought affected population constitutes >50% of that considering both climate change and population growth) in most regions (for example, North Australia, Southern Europe and Mediterranean, Central Europe, North Europe, West Coast South

-22-

          <Table 4, here, thanks>

**4 Discussions**

[revised manuscript text omitted]

Table 4: Changes in population exposure (including total, urban and rural population, mean ±
standard deviation of multi-model projections, unit: million) to severe drought at the globe and in
27 world regions in the 1.5°C and 2°C warmer worlds relative to the baseline period using the
fixed SSP1 population in 2100 (at the end of the 21$^{st}$ century). The italic numbers in the brackets
show the people affected solely by severe droughts under two future warming scenarios, which
were calculated using the fixed SSP1 baseline population in 2000 (do not consider the population

| Regions | 1.5°C warming | | | 2.0°C warming | | |
|---|---|---|---|---|---|---|
| | Total | Urban | Rural | Total | Urban | Rural |
| ALA | 0.0±0.1(-0.0±0.1) | 0.0±0.1(-0.0±0.1) | 0.0±0.0(0.0±0.0) | 0.0±0.1(0.0±0.1) | 0.0±0.1(0.0±0.1) | 0.0±0.0(0.0± |
| CGI | 0.1±0.1(0.0±0.1) | 0.0±0.1(0.0±0.0) | 0.0±0.0(0.0±0.0) | 0.1±0.1(0.0±0.1) | 0.0±0.1(0.0±0.0) | 0.0±0.0(0.0± |
| WNA | 0.2±1.1(0.2±1.5) | 0.2±1.1(0.2±0.6) | 0.0±0.0(0.0±0.9) | 0.8±1.8(1.1±2.3) | 0.8±1.7(0.6±0.8) | 0.0±0.1(0.5± |
| CNA | 4.9±10.9(3.1±6.8) | 4.5±10.2(2.2±5.0) | 0.4±0.7(0.9±1.8) | 6.1±6.6(3.8±4.1) | 5.6±6.2(2.7±3.0) | 0.5±0.4(1.1± |
| ENA | 3.0±17.0(1.9±10.0) | 3.0±16.8(1.3±8.2) | 0.1±0.2(0.6±2.0) | 7.5±9.9(3.4±4.8) | 7.4±9.8(0.2±1.5) | 0.1±0.1(0.1± |
| CAM | 2.3±7.8(2.1±7.2) | 2.2±7.6(0.6±4.0) | 0.1±0.2(1.5±3.3) | 8.5±9.4(7.8±8.6) | 8.2±9.12(3.5±4.6) | 0.3±0.3(4.3± |
| AMZ | 2.1±2.0(2.0±2.1) | 1.9±1.9(0.9±1.1) | 0.2±0.2(1.1±1.1) | 4.6±4.3(4.8±4.5) | 4.1±3.9(1.6±1.8) | 0.5±0.6(3.2± |
| NEB | 1.4±2.3(2.0±3.1) | 1.3±2.1(0.6±1.0) | 0.1±0.2(1.4±2.1) | 3.8±3.6(5.3±4.9) | 3.4±3.3(1.4±1.5) | 0.4±0.3(3.9± |
| WSA | 4.2±15.1(2.3±3.4) | 4.2±14.1(1.9±2.8) | 0.1±1.1(0.5±0.7) | 13.0±17.9(2.4±2.7) | 12.5±16.7(1.8±2.2) | 0.5±1.3(0.6± |
| SSA | 2.5±4.1(2.6±4.3) | 2.5±4.1(1.5±2.7) | 0.0±0.1(1.2±1.9) | 3.5±3.8(3.9±4.3) | 3.4±3.7(1.9±2.1) | 0.1±0.1(2.1± |
| NEU | 3.0±4.7(2.4±3.7) | 3.0±4.5(1.5±2.2) | 0.1±0.2(0.9±1.5) | 4.3±9.7(3.5±7.7) | 4.2±9.5(2.2±4.8) | 0.1±0.3(1.3± |
| CEU | 10.5±8.7(12.0±10.3) | 10.1±8.4(6.4±5.7) | 0.4±0.3 (5.5±4.7) | 18.4±24.0(22.6±28.2) | 17.7±23.0(11.5±14.9) | 0.8±1.0(11.1± |
| MED | 9.8±6.5(9.0±6.0) | 9.2±6.1(3.6±2.2) | 0.5±0.5(5.4±4.1) | 22.5±17.8(21.1±16.5) | 21.2±16.7(8.1±6.6) | 1.3±1.1(13.0± |
| SAH | -0.3±7.0(0.2±3.6) | 0.0±5.5(0.5±1.6) | -0.3±1.6(-0.3±2.5) | 1.7±9.1(1.4±4.7) | 1.8±7.2(1.3±2.3) | -0.2±2.1(0.1± |
| WAF | 2.7±71.7(1.9±22.5) | 2.9±66.3(0.7±3.2) | -0.2±5.5(1.2±19.5) | 14.3±109.5(6.3±34.6) | 14.0±101.2(1.9±5.1) | 0.3±8.4(4.4± |
| EAF | -23.7±58.0(-9.2±23.5) | -20.7±51.1(-1.4±3.1) | -3.0±7.0(-7.8±20.6) | -26.3±63.8(-10.2±26.3) | -22.6±56.2(-1.8±3.0) | -3.7±7.7(-8.4 |
| SAF | 5.8±4.2(3.2±2.1) | 5.2±3.8(1.1±0.6) | 0.6±0.5(2.1±1.8) | 12.0±9.0(7.3±5.0) | 11.0±8.2(2.9±1.5) | 1.0±0.8(4.4± |
| NAS | 1.1±4.1(1.7±6.3) | 1.1±3.7(0.8±3.3) | 0.0±0.4(0.9±3.0) | 1.1±5.1(1.7±7.8) | 1.0±4.6(0.6±4.1) | 0.1±0.5(1.1± |
| WAS | 4.2±15.1(4.7±10.4) | 4.2±14.1(1.5±3.0) | 0.1±1.1(3.2±7.6) | 13.0±17.9(11.8±12.0) | 12.5±16.7(3.8±3.5) | 0.5±1.3(8.0± |
| CAS | 1.7±15.3(1.6±10.6) | 1.0±14.0(0.5±4.7) | 0.7±1.5(1.1±6.1) | 7.0±9.4(5.3±6.6) | 5.5±8.8(1.7±3.2) | 1.5±1.0(3.6± |
| TIB | 0.1±3.2(0.3±2.6) | 0.0±2.8(-0.0±1.2) | 0.1±0.6(0.3±2.5) | 0.6±2.9(1.0±3.4) | 0.4±2.6(0.2±1.2) | 0.2±0.7(0.8± |
| EAS | 17.3±17.8(31.6±34.1) | 16.6±16.2(16.4±15.6) | 0.7±1.8(15.3±19.9) | 24.8±32.8(46.2±64.2) | 24.0±30.0(22.8±26.0) | 0.9±3.0(23.4 |
| SAS | -15.9±68.8(-12.8±62.4) | -14.9±61.3(-7.7±34.9) | -1.0±8.1(-5.2±31.0) | -35.9±52.6(-31.9±47.7) | -32.3±47.1(-18.3±27.6) | -3.6±6.0(-13. |

growth with the development of socioeconomics in the future)

-40-

[revised manuscript text omitted]

---

## Referee Report (RR1)

This major revised article takes into account the majority of my remarks. I think that the methodology is now clear and the results are always interesting and better highlighted. I suggest publishing this paper in the ESD with technical correction. The details are available below.

**Major comments response**

For my previous major comments, the asked clarifications are done. The methodology is given clearly and that is understandable. The difference of the model could be better used in the explication of the results but that can be consider as sufficient in the current version of the paper because the main focus is on the impacted population. For my major one remark (justification of the number of models) and my second remark (classification), your responses and corrections are sufficient.

For the bias correction, I read your justification and I am not completely convinced. I think that is ok for the majority of the area but in some locations, the non-use of the bias corrections is problematic. When some of the CMIP5 models do not simulate the correct precipitation in the current climate, the analyze by comparison between the future and the present cannot be correct. However, you are right that we must have several data to correct the output of the model and that is difficult to correct it. I would have preferred to have with a model a quantification of the impact of the correction for some regions (Sahel). This correction could already be made only on a limited number of inputs. However, I find your justification and openness in your discussion/conclusion sufficient for this paper. One remark, with some bias correction methods, the correlation between the outputs are conserved after the correction (cf CDFt method, Michelangeli P, Vrac M, Loukos H (2009)).

**Some questions/remarks**

I agree with all the changes.

**2 new remarks**

In your abstract, you show results (that is nice) but as you show only the urban and total impacted people that is not clear in the abstract why the number of people is smaller for the total.

Line 152: extra space after "field"